# Ethnic disparities in COVID-19 mortality and cardiovascular disease in England and Wales between 2020-2022

Marta Pineda-Moncusí [1], Freya Allery [2,3], Hoda Abbasizanjani [4], David Powell[4], Albert Prats-Uribe [1], Johan H. Thygesen [2], Angela Wood [5,6], Christopher Tomlinson [2], Amitava Banerjee [2], Ashley Akbari [4], Antonella Delmestri [1], Laura C. Coates [1], Spiros Denaxas[2,6], Kamlesh Khunti[7], Gary Collins [1], Daniel Prieto-Alhambra [1,8] & Sara Khalid [1] ✉ On behalf of the CVD-COVID-UK/COVID-IMPACT Consortium*

An increased risk of COVID-19 mortality risk among certain ethnic groups is well-reported, however data on ethnic disparities in COVID-19-related cardio-vascular disease (CVD) are lacking. We estimated age-standardised incidence rates and adjusted hazard ratios for 28-day mortality and 30-day CVD by sex for individual ethnicity groups from England and Wales, using linked health and administrative data. We studied 6-level census-based ethnicity group classification, 10-level classification (only for Wales), and 19-level classification as well as any ethnicity sub-groups comprising >1000 individuals each (only for England). COVID-19 28-day mortality and 30-day CVD risk was increased in most non-White ethnic groups in England, and Asian population in Wales, between 23rd January 2020 and 1st April 2022. English data show mortality decreased during the Omicron variant's dominance, whilst CVD risk [95% confidence interval] remained elevated for certain ethnic groups when compared to White populations (January-April 2022): by 120% [28-280%] in White and Asian men and 58% [32-90%] in Pakistan men, as compared to White British men; and by 75% [13-172%] in Bangladeshi women, 55% [19-102%] in Caribbean women, and 82% [31-153%] in Any Other Ethnic Group women, as compared to White British women. Ethnically diverse populations in the UK remained disproportionately affected by CVD throughout and beyond the COVID-19 pandemic.

Health inequity is multifaceted and often underpinned by a complex interplay of determinants, including but not limited to race and ethnicity, sex, and socioeconomic status. Underlying disparities were particularly exacerbated by and highlighted during the COVID-19 pandemic, where people from ethnically diverse backgrounds were disproportionately affected[1–6].

In the United Kingdom (UK), Asian, Black, and those from Mixed ethnic backgrounds were found to have higher COVID-19 mortality than the majority White population[7]. The Office for National Statistics (ONS) in the UK reported a higher mortality rate in the Chinese ethnic group during the first wave compared to the second, whereas the opposite was observed in those of Pakistani ethnicity[8,9]. Likewise,

A full list of affiliations appears at the end of the paper. *A list of authors and their affiliations appears at the end of the paper.
✉ e-mail: sara.khalid@ndorms.ox.ac.uk

peri- and post-pandemic health inequalities across different ethnic sub-groups were exacerbated and have been established as a major public health concern[9,10]. Despite this, ethnic sub-groups are traditionally grouped together, thereby rendering an appropriate assessment and identification of the needs of diverse communities as an unmet need. On the other hand, prior studies have associated COVID-19 with numerous cardiac complications[11], and an increased mortality risk in people with a history of cardiovascular disease (CVD)[12]. The ONS reported higher CVD mortality in ethnic minorities[13], however, little is known about the differences in CVD across ethnic groups after COVID-19 infection[14].

In this paper, we studied ethnic disparity in relation to severe outcomes after COVID-19 in the English and Welsh population. We investigated the risk of severe outcomes of COVID-19, including mortality and cardiovascular disease (CVD) among patients from different ethnic backgrounds and during different phases of the COVID-19 pandemic, whilst accounting for a set of available social determinants. We explored ethnicity using both traditionally broader groups and sub-groups, as well as previously unreported and more specific sub-groups with more granular ethnicity classifications.

## Results

### Ethnic diversity in individuals diagnosed with COVID-19 in England and Wales

We identified 4,867,595 (60% women) individuals in England (Supplementary Fig. 1) and 451,077 (55% women) in Wales (Supplementary Fig. 2) who were registered in a General Practice for at least one year, were aged 30 years or older, and had a confirmed record of COVID-19 diagnosis. In England, the high-level ethnicity distribution was as follows: White (83.0%), Asian/Asian British (8.0%), Black/Black British (3.2%), Mixed (1.3%), and Other Ethnic Group (1.3%). In contrast, Wales was less diverse, where the distribution was: White (92.1%), Asian/Asian Welsh (2.5%), Black/Black Welsh (0.7%), Mixed (0.7%) and Other Ethnic Group (0.8%). Ethnicity was not reported for 3.3% of the individuals, in both England and Wales (Supplementary Table 3).

Ethnicity in the Welsh population was also available in 10 NER ethnic groups classification (Supplementary Table 4), whilst the larger size of the English population permitted us to study ethnicity more granularly, including the 19 NHS ethnicity codes and SNOMED-CT concepts. Baseline characteristics for the high-level ethnic groups and their corresponding 19 sub-groups in England are reported in Supplementary Table 5 and Supplementary Table 6, respectively.

Mean age at diagnosis of COVID-19 infection in England ranged from 43.67 years (standard deviation [SD]: 12.56) in the Bangladeshi to 55.70 years (SD: 16.80) in the Irish populations (Supplementary Table 6); whilst in Wales, it ranged from 43.55 (SD: 11.57) in the Mixed group to 54.56 (SD: 16.38) in the Unknown group (Supplementary Table 4).

In England, those with Pakistani ethnicity had the highest proportion of individuals living in the most deprived areas (46.8% in the lowest index of multiple deprivation [IMD] fifth), followed by Bangladeshi (41.2%), African (39.5%), Arab (36.8%), Caribbean (35.3%), Any other Black background (35.1%), White and Black Caribbean (32.9%), White and Black African (32.2%), which were well above White British (19.6%), Indian (17.5%) and Chinese (17.2%) populations (Supplementary Table 6). In Wales, the ethnic group with the highest proportion of individuals from the most deprived areas was Black African (46.9%), followed by Pakistani (38.0%), Black Caribbean (37.1%), Other Ethnic Group (33.7%), Bangladeshi (31.5%) and Mixed (31.4%) (Supplementary Table 4).

### Incidence and hazard ratio differences in COVID-19 mortality and CVD between and within ethnicity groups

Due to the low number of individuals from non-White groups in the Welsh population and their lower number of outcome events

observed, we report estimates for Wales using the 6-level ethnicity categorisation. More granular results for Wales are presented in Supplementary Table 7 and Supplementary Fig. 4.

### Incidence rates [IR]

**28-day COVID-19 mortality (age-standardised IR [95%CI] per 100,000 population/year).** In England, all non-White ethnic groups, except those with missing ethnicity, had higher incidence of mortality than White (Fig. 1A). Conversely, those with Unknown ethnicity had the highest mortality rates in Wales, and only Asian/Asian British group, and men self-identified as Black/Black British and Mixed show a significant incremented age-standardised IR compared to White population (Fig. 1C).

A larger disparity in mortality rates is observed on the 19-level ethnicity group classification than using the broader categories in the English population, such as within the Asian/Asian British population, where mortality incidence in Bangladeshi (men: 116.8 [106.9 to 126.6], women: 65.5 [58.5 to 72.4]) were higher than Pakistani (men: 81.3 [77.0 to 85.6], women: 49.7 [46.4 to 52.9]) and Indian (men: 64.6 [61.5 to 67.7], women: 39.5 [37.1 to 42.0]).

At the most granular level of ethnicity classification available (SNOMED-CT classification, available only in England), we observed large differences between Central/South/Latin American (men: 178.5 [153.7 to 206.2], women: 55.2 [42.1 to 71.3]) and Iranian (men: 33.4 [23.4 to 46.2], women 10.2 [5.5 to 18.4]) within Other Ethnic Group; and between Nigerian and Somali men (137.0 [115.9 to 162.0] and 89.9 [72.8 to 110.0], respectively).

**30-day CVD (age-standardised IR [95%CI] per 100,000 population/year).** Four ethnic groups of the high-level classification in England (i.e., Asian/Asian British, Mixed, Black/Black British, and Other Ethnic Group) were more likely to experience a CVD than the White group (Fig. 1B). In Wales, confidence intervals were wide for ethnic groups other than White, thus, only Asian/Asian British women show an incremented incidence (Fig. 1C).

Consistent with mortality incidences, the 19-level ethnic group classification in England showed different CVD incidence within the high-level ethnic group classification, such as higher CVD incidence in Pakistani (men: 85.03 [80.98 to 89.08], women: 39.2 [36.5 to 41.9]) and Bangladeshi (men: 88.74 [81.12 to 96.37], women: 38.3 [33.5 to 43.1]) sub-groups vs the other Asian sub-groups.

At the most granular level, SNOMED-CT ethnicity concepts revealed a larger incidence among Turkish/Turkish Cypriot (men: 93.2 [75.3 to 111.1], women: 44.2 [32.5 to 58.7]) and among "Middle Eastern" women (excluding Israeli, Iranian and Arab, age-standardised IR: 73.5 [58.1 to 91.8]), compared with their corresponding ethnic group in high-level classification (i.e., Other Ethnic Group).

Supplementary Table 8 summarises all age-standardised IR estimates for 28-day mortality and 30-day CVD for England, whilst Supplementary Table 7 summarises it for Wales.

**Hazard Ratios [HR] (with White as reference group).** Survival analyses for England showed that the differences in mortality (Fig. 2A) and CVD (Fig. 2B) observed in age-standardised IR were maintained even when adjusted by age, IMD, vaccination status, pregnancy, geographical location in England, time of diagnosis, comorbidities, and medication/s use. In Wales, increased risk of mortality was confirmed in Asian/Asian British and Unknown, and increased CVD risk in Asian women and men with unknown ethnicity (Fig. 3).

Supplementary Table 9 and Supplementary Table 10 summarises all HR estimates for 28-day mortality and 30-day CVD, and their adjustments, respectively for England. Supplementary Table 11 and Supplementary Table 12 have the HR estimates for 28-day mortality and 30-day CVD, and their adjustments, respectively for Wales.

**A) Incidence rates of 28-day COVID-19 mortality in England:**

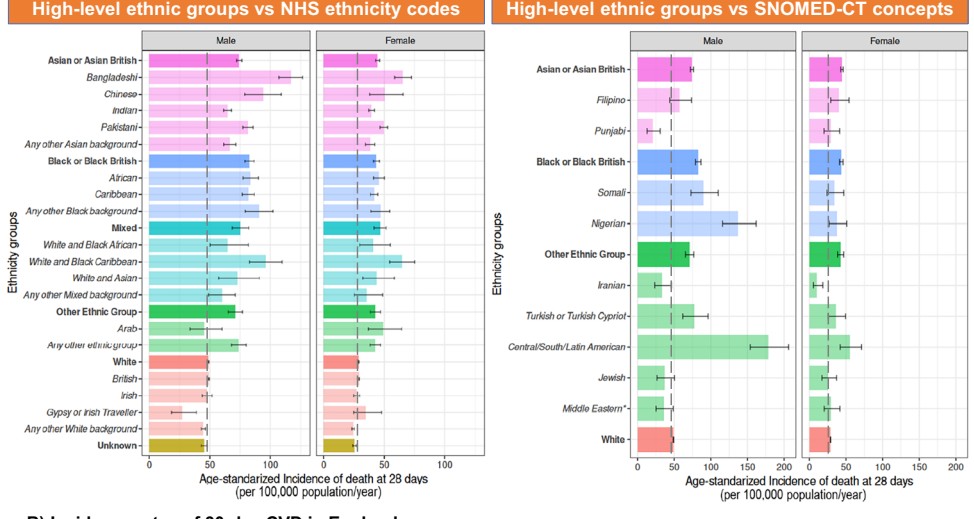

**B) Incidence rates of 30-day CVD in England:**

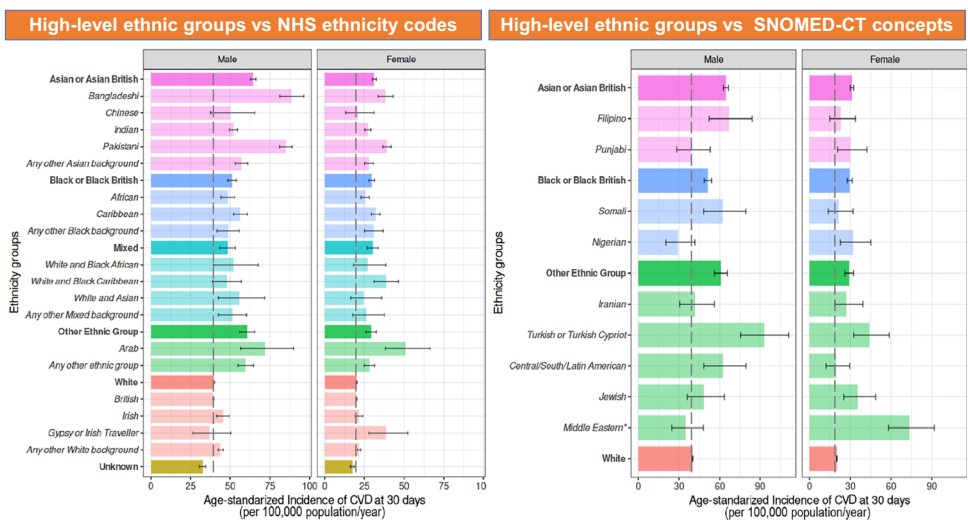

**C) Incidence rates of 28-day COVID-19 mortality and 30-day CVD in Wales:**

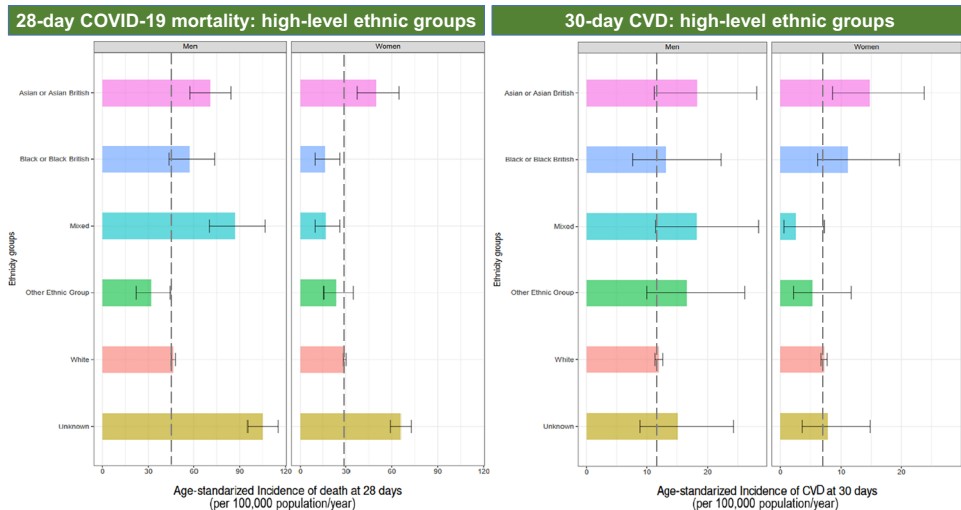

## Incidence and hazard ratio of COVID-19 mortality and CVD across and within ethnicity groups: trend over time

Reduced number of outcomes in Wales impairs their reliability when stratified over time. Thus, the following estimates reporting incidence and survival over time are focused on England population. Wales IR

estimates over time are included in Supplementary Table 13 and Supplementary Fig. 5.

**Incidence rates in England.** Supplementary Table 14 summarises England's age-standardised IR estimates for 28-day mortality and 30-day

**Fig. 1 | Age-standardised incidence rates (per 100,000 population/year) of 28-day mortality and 30-day CVD in England and Wales.** A 28-Day mortality in England, **B** 30-day CVD in England, and C 28-day mortality and 30-day CVD in Wales, among COVID-19 patients aged ≥30 years and stratifying by ethnicity group. Dark colours (and denoted in bold in the *Y* axis) represent the 6 high-level groups, and light colours (and denoted in italics in the *Y* axis) correspond to 19 NHS ethnicity codes sub-categories or SNOMED-CT concepts in England, and to the 10 ethnic groups in Wales. A vertical black dashed line marks the estimates from the White high-level group. To estimate the age-standardised incidence rates, age-specific incidence rates were calculated for 5-year age bands and then combined using the 2013 European Standard Population weights from 30 to 90+ age groups. Estimates are reported with their 95% confidence intervals. Explicit numerical values are available in Supplementary Table 7 for Wales and Supplementary Table 8 for England results. CVD cardiovascular disease, Middle eastern*, excluding Israeli, Iranian, and Arab.

## A) Risk of 28-day COVID-19 mortality by ethnicity in men (left panel) and women (right panel) range 30 to 100 years in England:

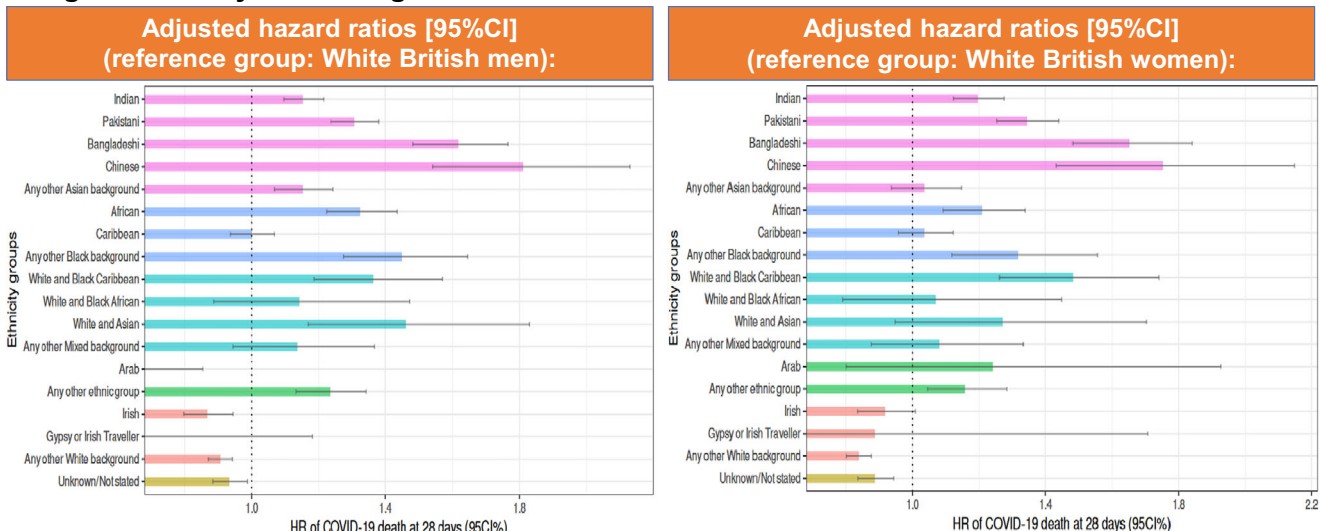

## B) Risk of 30-day CVD by ethnicity in men (left panel) and women (right panel) range 30 to 100 years in England:

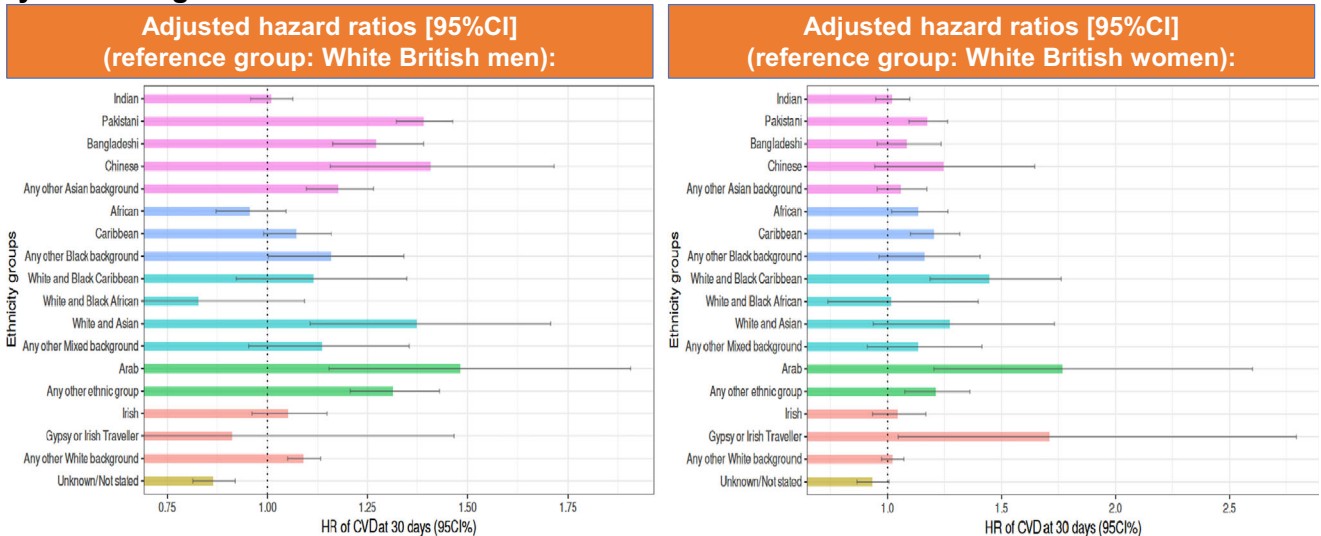

**Fig. 2 | Adjusted hazard ratios of 28-day mortality and 30-day CVD in England.** **A** 28-Day mortality and **B** 30-day CVD from individuals diagnosed with COVID-19 with diverse ethnic background in England, using White British ethnicity as the reference group. Dot lines in 1 highlight the risk from the reference group. Models were adjusted by age, ethnicity, deprivation index, vaccination status, geographic location in England, period of recorded COVID-19 diagnosis and comorbidities. Displayed hazard ratios belong to ethnicity coefficients and are reported with their 95% confidence intervals. Explicit numerical values are available in Supplementary Table 9 for 28-day mortality and Supplementary Table 10 for 30-day CVD results. CVD cardiovascular disease, CI confidence intervals, HR hazard ratios.

CVD over time. Population size for ethnic groups observed through the SNOMED-CT concepts were too small to be explored across time.

**28-day mortality (age-standardised IR [95%CI] per 100,000 population/year).** There was an overall decrease in mortality incidence from January 2020 to June 2022. At the beginning (from January to June 2020), the age-standardised IR for non-White ethnic groups (except those with Unknown ethnicity) were higher than the White group. This difference (with respect to the White group) fluctuated in magnitude but remained during the subsequent 18 months, disappearing only in the final 6 months until April 2022 (Fig. 4).

**A) Risk of 28-day COVID-19 mortality by ethnicity in men (left panel) and women (right panel) range 30 to 100 years in Wales:**

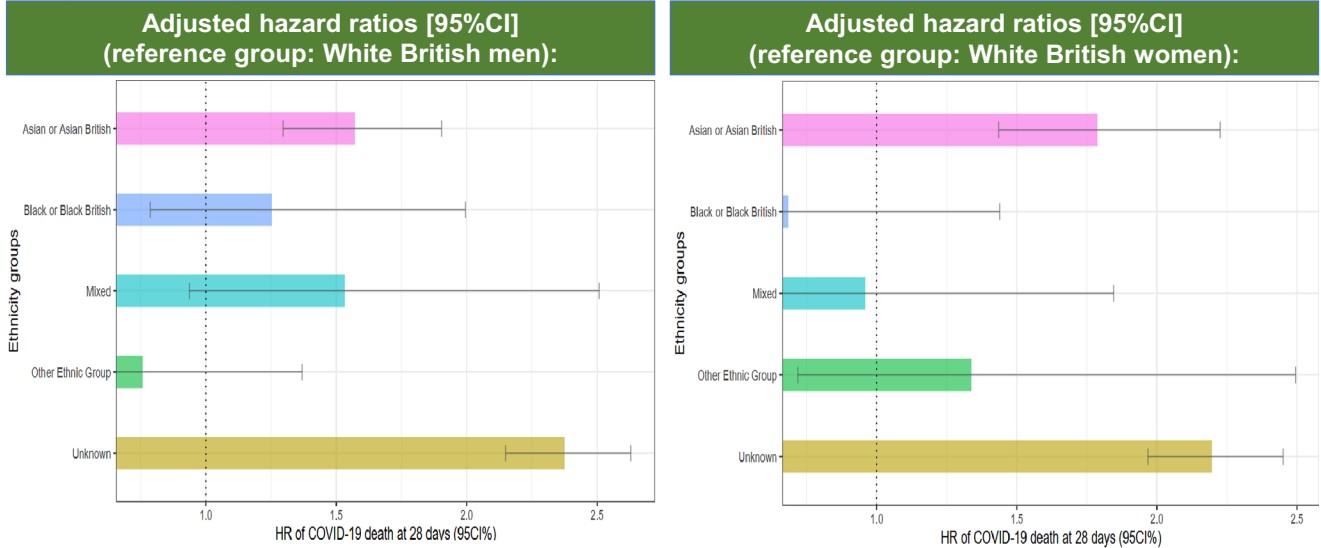

**B) Risk of 30-day CVD by ethnicity in men (left panel) and women (right panel) range 30 to 100 years in Wales:**

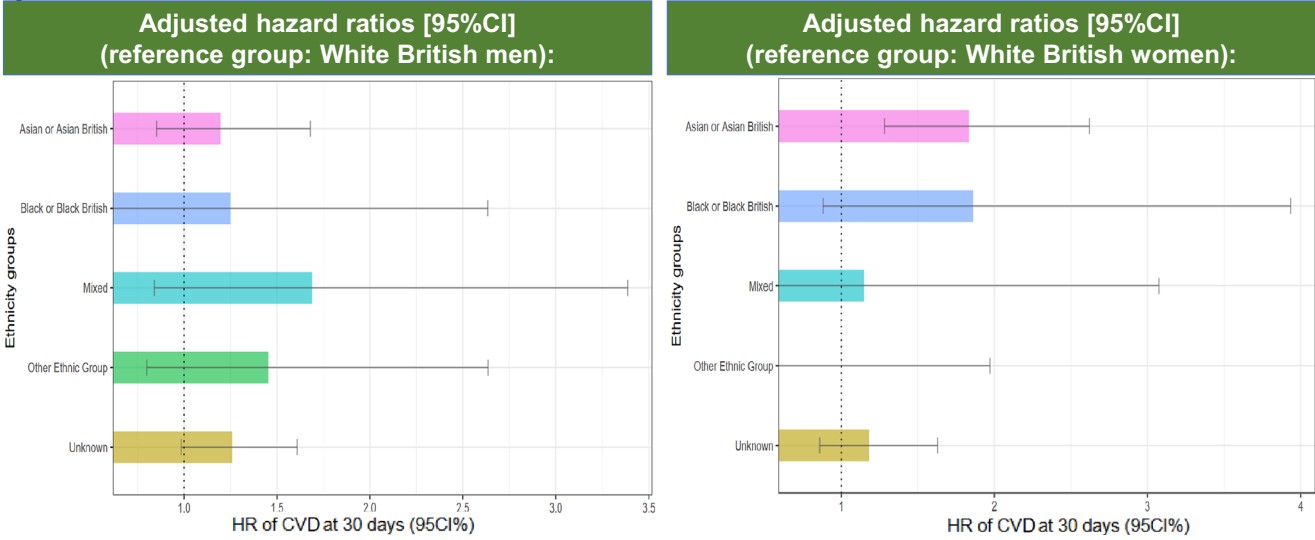

**Fig. 3 | Adjusted hazard ratios of 28-day mortality and 30-day CVD in Wales. A** 28-Day mortality and **B** 30-day CVD from individuals diagnosed with COVID-19 with diverse ethnic backgrounds in Wales, using White British ethnicity as the reference group. Dot lines in 1 highlight the risk from the reference group. Models were adjusted by age, ethnicity, deprivation index, vaccination status, geographic location in England, period of recorded COVID-19 diagnosis, and comorbidities. Displayed hazard ratios belong to ethnicity coefficients and are reported with their 95% confidence intervals. Explicit numerical values are available in Supplementary Table 11 for 28-day mortality and Supplementary Table 12 for 30-day CVD results. CVD cardiovascular disease, CI confidence intervals, HR hazard ratios.

The 19 sub-groups display better the fluctuations in the mortality rates across the different ethnic sub-groups, where Bangladeshi and Pakistani populations stand out by always having higher mortality rates than White British during the first two years of the pandemic (i.e., from 23rd January 2020 until 31st December 2021). Despite the overall decrease and the closed mortality gap for most of ethnic groups in the last study period (1st January to 1st April 2022), Pakistani men still presented an incremented incidence (14.97 [8.74 to 24.08]) compared to with British (5.83 [5.42 to 6.23]).

**30-day CVD (age-standardised IR [95%CI] per 100,000 population/year).** Age-standardised IR of CVD were generally higher during the first 6 months of the pandemic (i.e., 01st January to 30th June 2020), similar from July 2020 to June 2021, and slightly lower after July 2021. Likewise, with mortality, inequities in CVD incidence (with respect to the White group) varied over the studied period. However, the gap between distinct non-White groups compared to the White British was maintained in the final 6 months until April 2022 (Fig. 5).

Fluctuations in CVD rates can be better represented through the 19 sub-groups. Within Arian/Asian British, Bangladeshi and Pakistani populations constantly emerged as presenting incremented CVD rates when compared to White British over time, whilst the Chinese population was not significant incremented in the final 6 months (i.e., 1st January to 1st April 2022). Within Black/ Black British, Caribbean women presented continuous incremented rates during the full study period, and Caribbean men after the initial 12 months (i.e., 1st January 2021 to 1st April 2022), whilst rates in African women were only incremented at the initial 6 months and rates in African men were only incremented at the final 6 months.

## A) 28-day COVID-19 mortality in men:

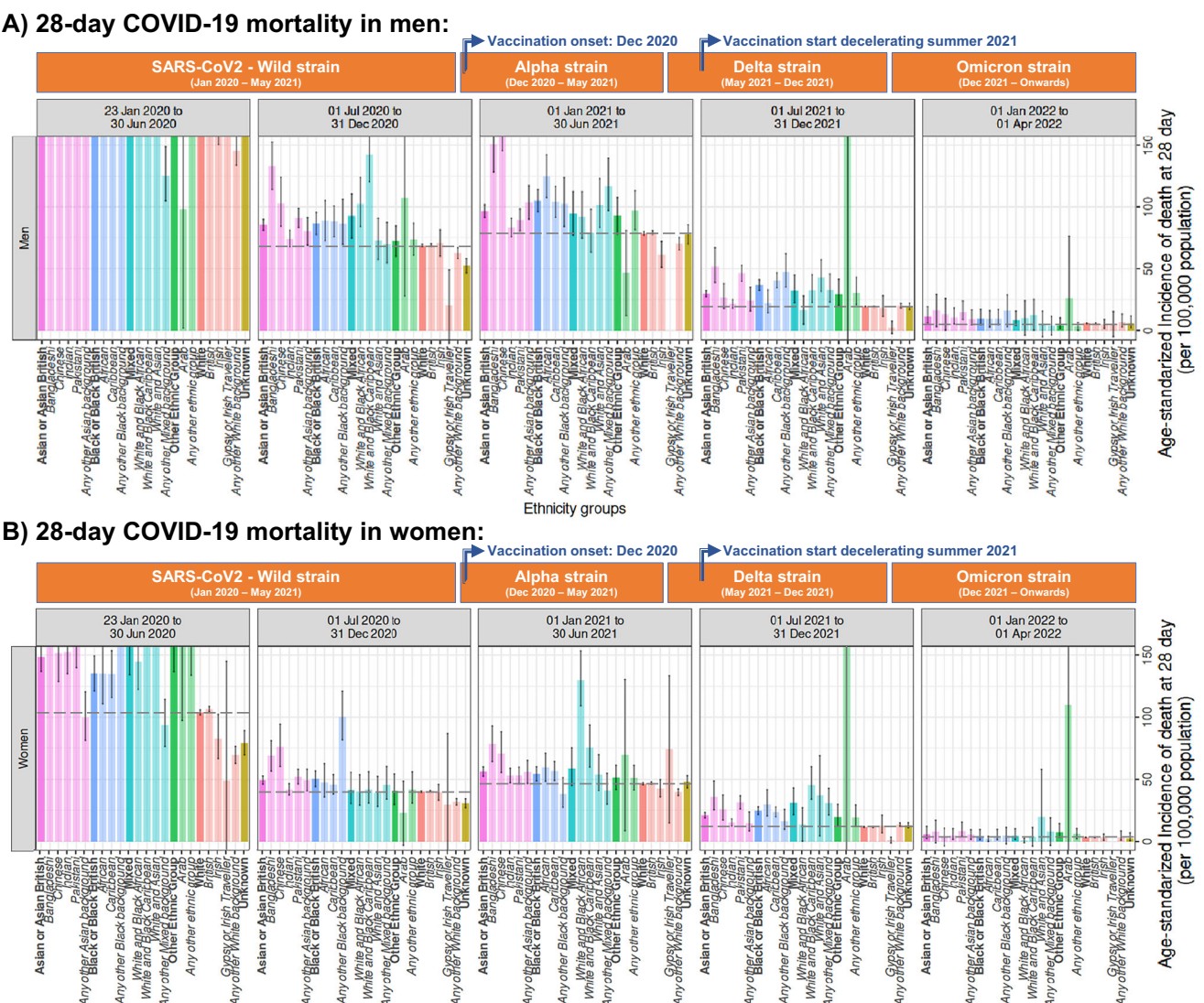

## B) 28-day COVID-19 mortality in women:

**Fig. 4 | Age-standardised incidence rates of 28-day mortality (per 100,000 population/year) by period of recorded COVID-19 diagnosis in England. A** Men and **B** women diagnosed with COVID-19 between 30 and 100 years old and across different ethnic groups in England. Dark colours represent the 6 high-level groups and light colours the corresponding 19 NHS ethnicity codes or SNOMED-CT concepts, which are denoted in bold and italics, respectively, in the *X* axis. Dotted horizontal black lines mark the estimates from the White high-level group.

Important dates for contextualisation, such as the entrance of SARS-CoV2 variants and vaccination in the UK, have been included. To estimate the age-standardised incidence rates, age-specific incidence rates were calculated for 5-year age bands and then combined using the 2013 European Standard Population weights from 30 to 90+ age groups. Estimates are reported with their 95% confidence intervals. Explicit numerical values are available in Supplementary Table 14.

### Hazard Ratios (with White as reference group) in England

**28-day mortality.** During the first 6 months of the pandemic, women from Other Ethnic Group (HR [95%CI]: 1.31 [1.11 to 1.54]), and individuals from Asian/Asian British (HR [95%CI]: 1.19 [1.12 to 1.27] in men, 1.30 [1.19 to 1.42] in women) and Mixed (HR [95%CI]: 1.24 [1.05 to 1.45] in men, 1.24 [1.03 to 1.51] in women) had an increased risk of mortality post COVID-19 as compared to White population. Whilst those with an increased mortality in the last 6 months of the study period were women from Other Ethnic Group (HR [95%CI]: 2.06 [1.09 to 3.88] in women), and Asian/Asian British (HR [95%CI]: 1.40 [1.08 to 1.82] in men, 1.52 [1.08 to 2.12] in women) (Supplementary Fig. 6).

Considering the 19 ethnic sub-groups, mortality risk (Fig. 6) was increased in Bangladeshi and Pakistani from the onset of the pandemic until end of December 2021, where the lower HR [95%CI] of Bangladeshi men and women were 1.55 [1.33 to 1.81] and 1.47 [1.22 to 1.77], respectively, and risk estimates in Pakistani ranged from 1.15 [1.04 to

1.28] to 1.33 [1.15 to 1.55] in men and 1.16 [1.02 to 1.32] to 1.54 [1.29 to 1.84] in women. We observed other differences, such as an increased mortality during the first 12 months of the pandemic (from 23rd January to 31st December 2020) in Any Other Black background, and in men self-identified as Indian or White and Black Caribbean; in Indian women during the first 6 months, or in White and Black Caribbean women during the first 6 months and from 1st July to 31st December 2021.

No significant differences in 28-day mortality risk were detected after January 2022 (i.e., 1st January to 1st April 2022) (Supplementary Table 15).

**30-day CVD.** During the first 6 months, only women from Mixed (HR [95%CI]: 1.89 [1.43 to 2.49]) and Asian/Asian British (HR [95%CI]: 1.20 [1.04 to 1.38]) high-level groups had a significant increased risk of CVD compared to the White group. Whilst in the last 6 months increased

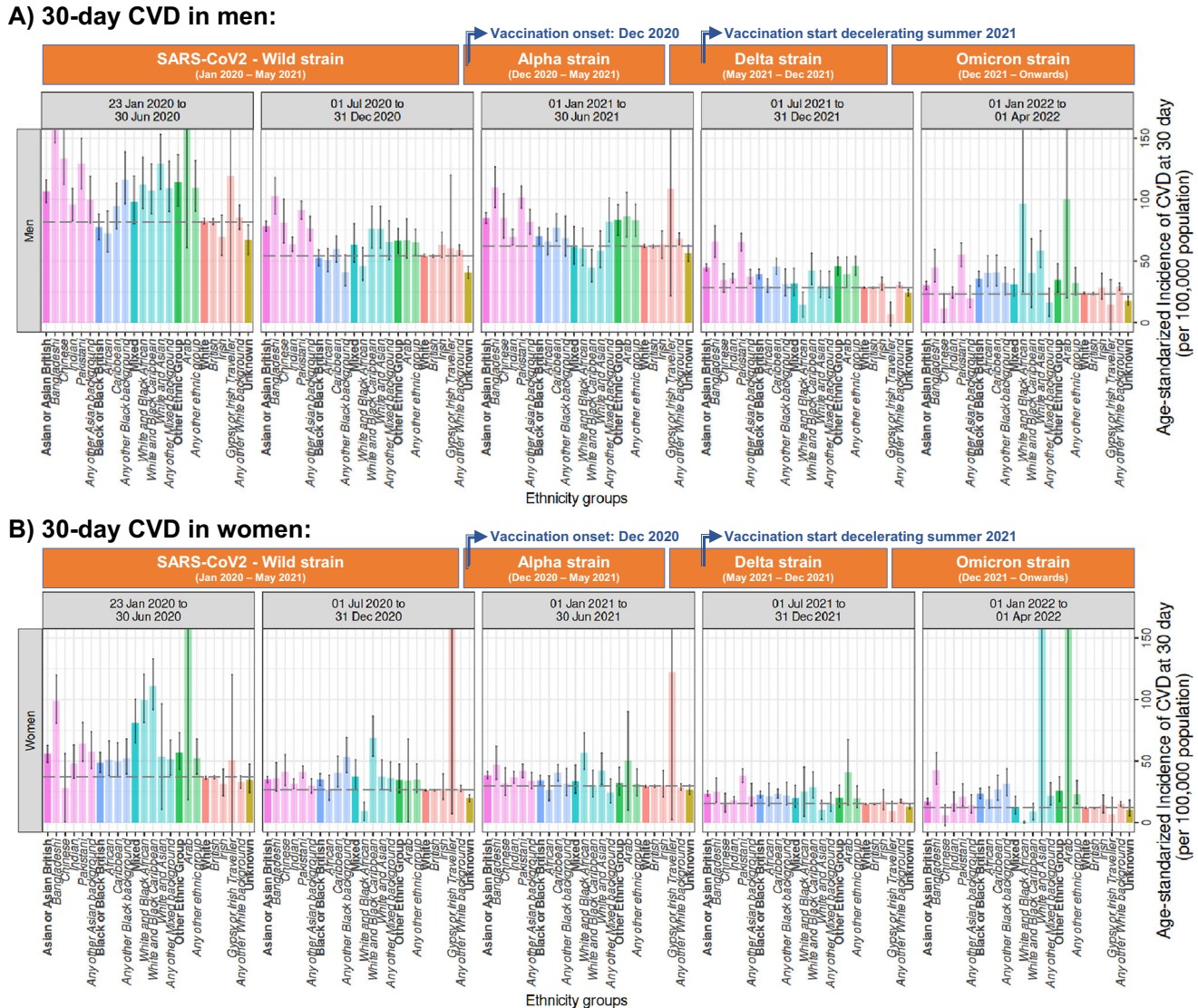

**Fig. 5 | Age-standardised incidence rates of 30-day CVD (per 100,000 population/year) by period of recorded COVID-19 diagnosis in England. A** Men and **B** women diagnosed with COVID-19 between 30 and 100 years old and across different ethnic groups in England. Dark colours represent the 6 high-level groups and light colours the corresponding 19 NHS ethnicity codes or SNOMED-CT concepts, which are denoted in bold and italics, respectively, in the *X* axis. Dotted horizontal black lines mark the estimates from the White high-level group. Important dates for contextualisation, such as the entrance of SARS-CoV2 variants and vaccination in the UK, have been included. To estimate the age-standardised incidence rates, age-specific incidence rates were calculated for 5-year age bands and then combined using the 2013 European Standard Population weights from 30 to 90+ age groups. Estimates are reported with their 95% confidence intervals. Explicit numerical values are available in Supplementary Table 14.

CVD risk was observed in women from Other Ethnic Group (HR [95% CI]: 1.83 [1.34 to 2.52]) and Black/Black British (HR [95%CI]: 1.39 [1.14 to 1.71]). Conversely, men from Asian/Asian British and Other Ethnic Groups show an increased CVD risk from July 2020 to December 2021 (Supplementary Fig. 7).

When observing the initial 6 months of the study using the 19 NHS ethnicity codes (Fig. 7), we could detect an increased CVD risk in Arab women (HR [95%CI]: 3.80 [1.42 to 10.18]), and we observed that only Bangladeshi women (HR [95%CI]: 1.86 [1.23 to 2.80]) within the Asian/ Asian British groups, and women from the White and Black Caribbean (HR [95%CI]: 2.46 [1.63 to 3.73]) and White and Black African (HR [95% CI]: 2.30 [1.30 to 4.06]) within the Mixed groups had an incremented CVD. When observing the male sub-groups during the period of July 2020 to December 2021, only Pakistani and Bangladeshi were increased within the Asian/Asian British, and Any other ethnic group within Other Ethnic Group.

As a highlight, in the last 6 months of the study, the following sub-groups had a remaining increased CVD risk: men from Pakistani (HR [95%CI]: 1.58 [1.32 to 1.90]), White and Asian (HR [95%CI]: 2.20 [1.28 to 3.80]) or Any other White background (HR [95%CI]: 1.17 [1.04 to 1.31]), and women from Bangladeshi (HR [95%CI]: 1.75 [1.13 to 2.72]), Caribbean (HR [95%CI]: 1.55 [1.19 to 2.02]), or Any Other Ethnic Group (HR [95%CI]: 1.82 [1.31 to 2.53]) (Supplementary Table 16).

## Discussion

Ethnicity is a social construct composed of various components, including physical appearance, race, culture, language, religion, nationality, and aspects of identity[15]. Each ethnicity group is composed of a group of individuals who identify themselves within a collective that shares similarities, which may include similar health outcomes. While the causes of risk differences for a specific health outcome have been shown to be multifactorial[16], we can still identify disparities

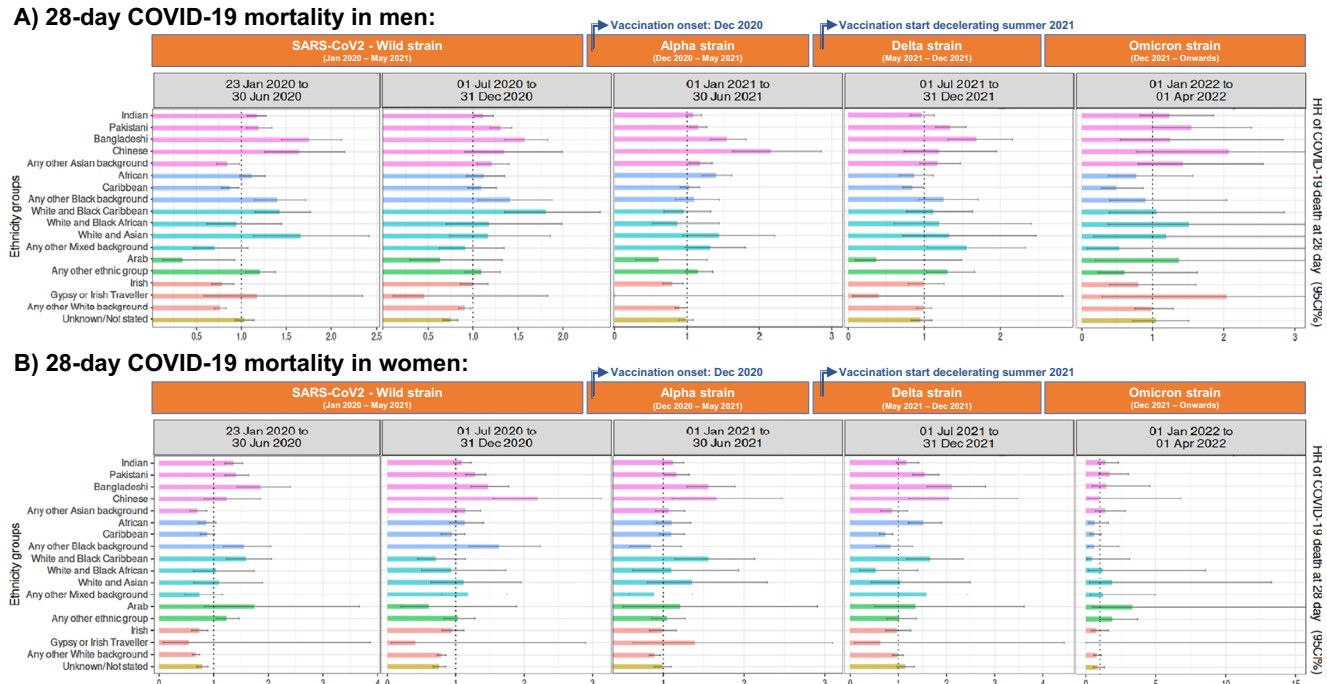

**Fig. 6 | Adjusted hazard ratios of 28-day mortality of the 19 NHS ethnicity groups by months of recorded COVID-19 diagnosis, using White ethnicity as the reference group, in England. A** Men and **B** women from England. Dot lines in 1 highlight the risk from the reference group. Models were adjusted by age, ethnicity, deprivation index, vaccination status, geographic location in England, period of recorded COVID-19 diagnosis and comorbidities. Displayed hazard ratios belong to ethnicity coefficients and are reported with their 95% confidence intervals. Explicit numerical values are available in Supplementary Table 15. CI confidence intervals, HR hazard ratios.

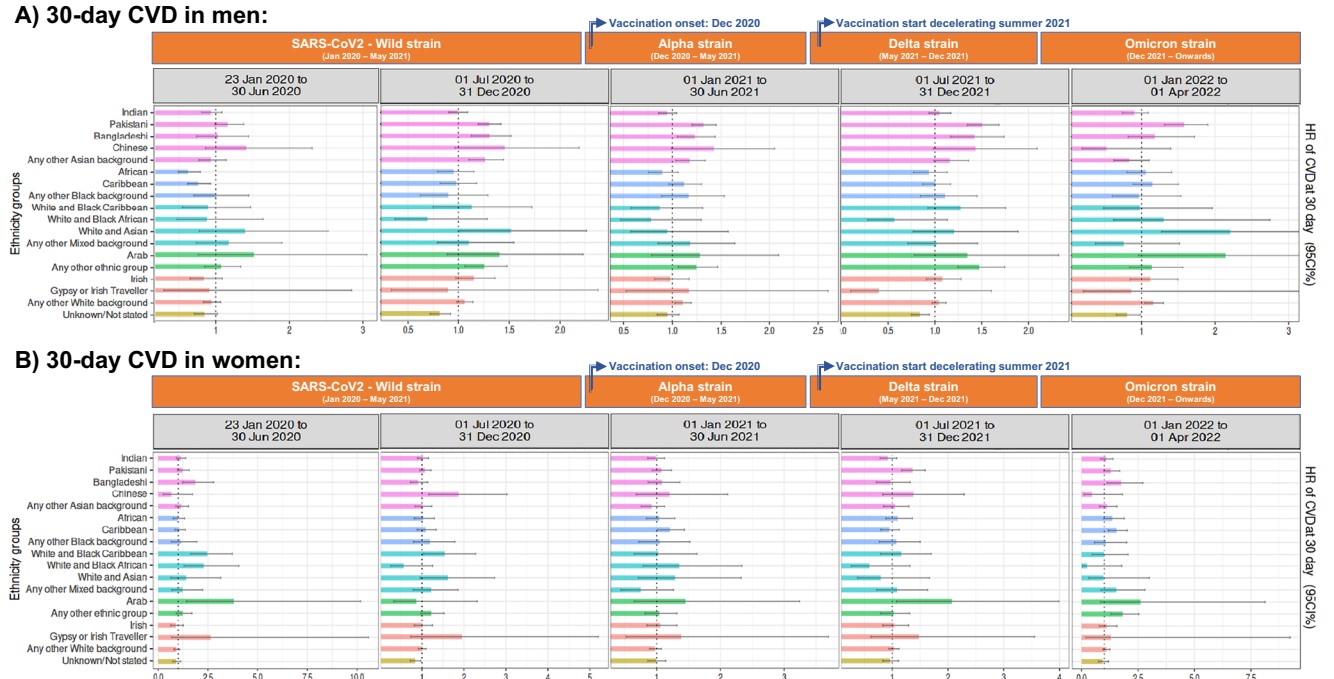

**Fig. 7 | Adjusted hazard ratios of 30-day CVD of the 19 NHS ethnicity groups by months of recorded COVID-19 diagnosis, using White British ethnicity as the reference group, in England. A** Men and **B** women from England. Dot lines in 1 highlight the risk from the reference group. Models were adjusted by age, ethnicity, deprivation index, vaccination status, geographic location in England, period of recorded COVID-19 diagnosis and comorbidities. Displayed hazard ratios belong to ethnicity coefficients and are reported with their 95% confidence intervals. Explicit numerical values are available in Supplementary Table 16. CVD cardiovascular disease, CI confidence intervals, HR hazard ratios.

across ethnic groups[17,18]. For instance, there are multiple studies in the UK showing that outcomes such as mortality, access to healthcare or quality of life were disproportionately affected across ethnic groups even before the COVID-19 pandemic[19–22]. When the pandemic struck, reports revealing that minority ethnic groups were more severely impacted by COVID-19 emerged not long after the initial months. To improve our understanding of the extent of the health inequities during the COVID-19 pandemic, this paper explored the ethnic diversity and health disparities of individuals aged ≥ 30 and ≤ 100 years who were diagnosed with COVID-19 between 23rd January 2020 to 1st April 2022 and registered with a primary care General Practice in England or Wales. We analysed their risk of mortality and CVD during the 2.5 years after the pandemic outbreak across different ethnic groups using both traditionally broader groups and sub-groups. Prior studies reported men had higher risk of COVID-19 mortality compared to women[3–6,23]. In addition, CVD is more prevalent in men than women in the general population[24]. Thus, we analysed the risk of COVID-19 mortality and CVD separately for men and women. Our results confirm men generally had higher incidence and risk of COVID-19 mortality than women, with fewer exceptions for specific ethnic groups where women's hazard rates were higher, such as Chinese women during the period of Jul-Dec 2020, or African and 'White and Black Caribbean' women during the period of Jul-Dec 2021. Likewise, men had higher incidence and risk of CVD, except for Middle Eastern women, whose incidence of CVD was higher than Middle Eastern men.

In line with previous estimates reported by the ONS[7], most of non-White ethnic groups had an increased mortality when compared with White populations in England and Wales. All increased estimates remained after adjusting for multiple confounders in England, and the increased risk of mortality was replicated in Asian/Asian British and Unknown ethnicities in Wales. Analysing the risk of mortality over time in England, our survival analysis showed the mortality gap across different ethnic groups disappeared after the appearance of the Omicron variant (period of Jan-April 2022)[25]. This could be a result of the vaccination combined with the lower severity of the original Omicron variant. Despite these findings, health disparities in COVID-19 outcomes were not over in the Omicron era when analysing CVD risk after SARS-CoV-2 infection.

Different ethnic groups in England and Wales had increased incidence of CVD after COVID-19 diagnosis, when compared to White population. The risk was confirmed after adjusting by confounders in England ethnic groups, and for Asian women and men with unknown ethnicity in Wales. However, and in contrast with mortality, CVD risk remained significantly increased for several ethnic groups (compared with White British) in England even after the emergence of Omicron. Several studies have highlighted how cardiovascular disease is a risk factor for COVID-19 infection severity[26,27], and how this risk is exacerbated among minority ethnic groups due to health inequities as well as the higher presence of certain comorbidities in specific communities[28]. However, little research has been done in terms of differences in cardiovascular risk post SARS-CoV-2 infection across distinct ethnic groups. One study, focused on US hospitals, reported higher rates of major adverse cardiovascular events (MACE: including death, myocardial infraction, stroke and heart failure) in hospitalised patients from the most vulnerable counties[14]. To our knowledge, this is the first time where CVD rates of individuals diagnosed with COVID-19 across a large number of different ethnic groups have been examined. The analysis we undertook shows where the gaps in COVID-19 mortality and CVD are and where healthcare providers can focus to address them.

## Strengths and limitations of this study
Our study fills a gap in the literature regarding the risk of CVD after SARS-CoV-2 infection across different ethnic groups. Use of highly granular ethnicity classifications had been recommended to capture important heterogeneity[29]. This can be challenging: the reduction of

the population size within these more detailed and specific ethnic groups usually leads to wider confidence intervals and therefore less certain estimates. I.e., there may not be sufficient statistical power to detect differences in the smallest groups. The lower number of individuals with a non-White ethnicity reported in Wales (i.e., n = 20,805 [4.6%]) and their low frequency of the study outcomes (between 0.4 and 5.3%), challenged the opportunities to make use of the Wales results using the 10-ethnicity groups classification. Welsh data was used to replicate English findings on less granular data. However, this study successfully reports differences in COVID-19 associated mortality and CVD across more than 19 ethnic groups, in England, and includes for first time data on more granular ethnic group categories, such as Iranian and Turkish communities, that have not previously been described. Use of more granular ethnicity classifications, such as the 19 NHS ethnic codes, illustrates the diversity of the study population that would be otherwise masked by broader classifications such as the 6 high-level ethnicity groups. However, as previously mentioned, the stratification into further smaller groups may compound the detection of any differences, even in a large population like England. Thus, it is possible that the number of ethnicities experiencing worse outcomes than the White population, and therefore facing health disparities, is even larger than reported both in England and Wales.

These results must be interpreted taking into account that the diagnosis of COVID-19 in these data did not include cases of lateral flow test (LFT)-only positive COVID-19 cases[30]. Moreover, due to limited capacity, testing was restricted outside of secondary care settings during the first wave, meaning diagnoses were clinical, resulting in an increased risk of potential misclassification during that period[31]. Despite the cohorts of the study might not include all new cases of COVID-19 in the UK during the study period, the data sources of the study have been previously considered representative of the populations in England and Wales[32,33]. The selected adjusting factors for COVID-19 mortality were obtained from the ONS[34], but instead of including severity measures (such as hospitalisation or intensive care unit admission) and specific medication for acute COVID-19, included health status before the COVID-19 diagnosis. Other factors may play a role in the interpretation of this results, such as health-seeking behaviour and barriers to accessing health care[35], which may exacerbate the differences between the White and non-White study population, where the observed non-White could be more populated by those who are experiencing worse outcomes.

To calculate our age-standardised IR, we replicated the ONS methodology, including the restriction of the population to ages from 30 to 100, but included 19 rather than 10 ethnic groups in the England population[36]. The age restriction was applied because the incidence of severe COVID-19 outcomes in individuals under 30 years of age was very rare, while individuals over 100 years were uncommon. In addition to the difference of how the study population was stratified into the ethnic groups, our study population might differ from the ONS. Since we have included individuals registered in the primary care systems from England and Wales, our study cohort may reflect more severely unwell individuals compared to the general population. Thus, mortality estimates show some differences across both studies (such as which group had the higher mortality incidence). However, the main finding that non-White British groups have a higher risk of mortality due to COVID-19 was consistent. Finally, we must acknowledge the likelihood that residual (unadjusted/unobserved) confounding could -at least partially- account for the observed differences, which is an inherent limitation of observational studies.

In conclusion, patients from non-White British population experienced worse mortality and cardiovascular outcomes after infection with SARS-CoV-2 during the 2.5 years after the pandemic outbreak. Risk fluctuated over the course of the pandemic, but Bangladeshi and Pakistani were the two ethnic groups where outcome

disparities were consistently increased over time. We confirmed a generally higher risk of COVID-19 mortality and CVD in men than in women. Ethnicity specific increased risk of mortality became non-significant after the entrance of Omicron variant, but risk of CVD remained increased for men with Pakistani, White and Asian or Any other White background, and women with Bangladeshi, Caribbean, or Any Other Ethnic Group. The reasons for these disparities are complex and intersectional, and further studies are required to explain this remaining increased risk of CVD for certain ethnic group of patients diagnosed from COVID-19. Our study results highly support targeted public health interventions as a means to reduce cardiovascular disease disparities after COVID-19. Further studies are recommended to monitor the ongoing disparities in CVD outcomes among COVID-19 patients from different ethnic groups.

## Methods

### Data Sources
**England.** We included deidentified data from eight linked data sources in the National Health Service (NHS) England Secure Data Environment (SDE) service for England, accessed via the BHF Data Science Centre's CVD-COVID-UK/COVID-IMPACT Consortium[32]. Linkage of individuals' records across these data sources was provided by the NHS England's Master Person Service using the NHS number, a unique 10-digit healthcare identifier[37]. The eight linked data sources are detailed in Table 1.

**Wales.** We used anonymised individual-level linked data held within the Secure Anonymised Information Linkage (SAIL) Databank at Swansea University[33], also accessed via the BHF Data Science Centre's CVD-COVID-UK/COVID-IMPACT Consortium. The data in SAIL are de-identified using multiple encryptions by different organisations. The linked data sources are detailed in Table 1[38,39].

### Data access
A data sharing agreement issued by NHS England enables approved researchers based in UK research organisations that co-sign this agreement to access the data held within the NHS England's SDE service for England. Data access was granted to the CVD-COVID-UK/COVID-IMPACT consortium via the NHS England's Online Data Access Request Service (ref: DARS-NIC-381078-Y9C5K).

All research conducted within the SAIL Databank trusted research environment (TRE) has been completed under the permission and approval of the SAIL independent Information Governance Review Panel (IGRP) project number 0911.

### Ethical approval
The North East - Newcastle and North Tyneside 2 research ethics committee provided ethical approval for the CVD-COVID-UK/COVID-IMPACT research programme (REC no: 20/NE/0161) to access, within secure trusted research environments, unconsented, whole-population, de-identified data from electronic health records collected as part of patients' routine healthcare.

Our project (proposal CCU037, short title: Minimising bias in ethnicity data) agreed the objectives of the consortium's ethical and regulatory approvals and was authorised by the BHF Data Science Centre's Approvals and Oversight Board. Approved researchers (M.P.M., F.A. and S.K.) conducted the analyses within the NHS England's SDE via secure remote access. Ensuring the anonymity of individuals, only summarised-aggregated results that were manually reviewed by the NHS England 'safe outputs' escrow service were exported from the SDE.

### Study design and duration
This cohort study analysed electronic health record (EHR) data sources from 23 January 2020, the date of the first documented case of COVID-19 in the UK, to 29 June 2022 (end of data availability on England, thereafter the end of the study period)[40]. Individuals were included on the date of their COVID-19 diagnosis (hereafter index date) and followed through to the earliest of these events: (i) death, ii) CVD or (ii) end of study period.

To account for wave and variant-related variation in infection and outcome rates during the phases of the pandemic, analysis was also stratified into 6-monthly windows covering the study period (e.g., 23rd January 2020 to 31st June 2020, and every 6 months thereafter up to the end of the study period). Individuals were included until 1st April 2022, but the outcomes of interest were observed until the end of the study period.

### Study population
Individuals aged between 30 and 100 years registered with a general practice in England or Wales (identified using GDPPR for England and WLGP for Wales) with a first record of a confirmed COVID-19 diagnosis from 23rd January 2020 to 1st April 2022 were included. We identified confirmed COVID-19 diagnosis as a record of a diagnosis in primary or secondary care (GDPPR and/or HES APC for England, WLGP and/or PEDW for Wales), a positive PCR test (in SGSS for England and PATD for Wales), or a hospital admission due to COVID (inclusion of the individual into the CHESS dataset for England). The index date was the first recorded date of COVID-19 diagnosis.

We excluded individuals who were < 30 or >100 years of age (consistent with other studies)[36] at the time of COVID-19 diagnosis, had no confirmed diagnosis of COVID-19, had less than one year of clinical history records in GDPPR for England (Supplementary Fig. 1) and in WLGP for Wales (Supplementary Fig. 2) before index the date, had an invalid sex record, and/or had a date of death before the index date.

### Outcomes of the study
**Two outcomes were studied.**

28-day mortality: defined as any death within the 28 days from COVID-19 event (positive test, diagnosis, or hospital admission due to COVID-19)[41,42].

30-day CVD: defined as the occurrence of a cardiovascular disease acute event (CVD) within 30 days after the COVID-19 event (included conditions are reported in Supplementary Table 1). CVD was defined as per International Classification of Diseases 10th Revision (ICD-10) codes in hospital data, a Systematised Nomenclature of Medicine Clinical Terms (SNOMED-CT) concepts in GDPPR or Read V2 in WLGP.

### Ethnicity classifications
Ethnicity is a multifaceted concept that encompasses various components, including physical appearance, race, culture, language, religion, nationality, and aspects of identity[15]. Ethnicity was self-reported by patients at the GP or hospital, and the most recent record was used in England, as defined by Pineda-Moncusí et al. (2024)[15], and in Wales, as defined by Akbari et al. (2024)[39]. Four different ethnicity classifications were applied to study differences at varying granularity. Classifications from the least to the most granular (i.e., to the most detailed or specific) were defined as:

1. High-level ethnicity groups: Asian/Asian British, Black/African/Caribbean/Black British, Mixed, Other Ethnic Groups, Unknown, and White. These six categories are based on the high-level ethnicity classification adopted by the ONS[43].
2. NHS ethnicity codes (for England): 19 standard ethnicity categories defined in the NHS England Data Dictionary, available in GDPPR and HES-APC data sources[44].
3. SNOMED-CT concepts (for England): records in the GDPPR data source containing ethnicity concepts in a SNOMED-CT UK Edition[45]. SNOMED concepts selected to be explored in this study were those who had ≥1000 individuals.

**Table .1 | Linked data sources included in England and Wales databases**

| Database (Country – full name) | Type of Data | Acronym | Data sources full name |
|---|---|---|---|
| England – National Health Service (NHS) England Secure Data Environment (SDE) service for England | Primary care data | GDPPR | The General Practice Extraction Service (GPES) Data for Pandemic Planning and Research |
| | Prescribed medication | Primary Care Meds | Medicines dispensed in Primary Care, NHS Business Services Authority data |
| | Hospital admissions data | HES-APC[1] | Episode Statistics for Admitted Patient Care |
| | | SUS[2] | Secondary Uses Service |
| | Hospital data specific for individuals hospitalised with COVID-19 | CHESS | COVID-19 Hospitalisation in England Surveillance System |
| | National laboratory COVID-19 testing data | SGSS | Public Health England Second Generation Surveillance System |
| | COVID-19 vaccination | – | From COVID-19 Vaccination Status and Vaccine Adverse Reactions tables obtained from hospital hubs, local vaccine services and vaccination centres in England |
| | Mortality information from the ONS: | – | Civil Registration of Deaths. |
| Wales – Secure Anonymised Information Linkage (SAIL) Databank at Swansea University | Demographic and mortality data | C20 | Welsh COVID-19 e-cohort [38] |
| | Primary care data | WLGP | Welsh Longitudinal General Practice data |
| | Dispensed medication | WDDS | Welsh Dispensing DataSet |
| | Hospital admissions data | PEDW | Patient Episode Dataset for Wales |
| | National laboratory COVID-19 testing data | PATD | COVID-19 test results for Wales |
| | COVID-19 vaccination data | CVVD | COVID Vaccine Data |
| | Ethnicity data | – | The research ready population-scale ethnicity-spine in Wales [39] |

[1] HES-APC is the national dataset for hospital admissions and is obtained directly from the patients' clinical records.[2] SUS was built for purposes other than direct clinical management and thus complements and contains more information than HES-APC, such as administrative information.

4. The 10 ethnicity groups (only for Wales) proposed for use by the New and Emerging Respiratory Virus Threats Advisory Group (NERVTAG, referred to as NER hereafter)[39,46].

Supplementary Table 2 describes the mapping between NHS ethnicity codes to High-level ethnic groups. Supplementary Fig. 3 shows the three classification systems in the NHS England and the impact of using them.

**Covariates**

Demographic characteristics of study population included: age, sex, ethnic group, index of multiple deprivation (IMD)[47], smoking (only in England) and location within the nine English regions or Wales (Wales was considered as one region due to population size); all extracted at index date.

The IMD is calculated for each LSOA in England to rank the most deprived areas (rank 1) to the least deprived, and is based on a combination of seven indicators that cover a broad spectrum, including income, barriers to housing and services, education and skills, employment, health and disability, crime, and living environment[48,49]; and it is often categorised into fifths, with 1 denoting the most deprived and 5 the least deprived areas[47]. The IMD was treated as an ordinal variable by the model. Smoking status was defined as ever smoker versus no smoking record in England, smoker status was not extracted in Wales. The geographical location of individuals within the nine English regions was reported based on Lower-layer Super Output Area (LSOA) codes.

Clinical characteristics: pregnant at index date; record of the first dose of COVID-19 vaccination before index date, recorded diagnosis of CVD within the year before index date; and recorded diagnosis ever before index date of CVD, atrial fibrillation, alcohol problems, bipolar disorder, cancer, chronic kidney disease, chronic obstructive pulmonary disease, chronic mental health disorders (including depression, schizophrenia and bipolar disorder), dementia, diabetes, hypertension, obesity, osteoporosis, and rheumatoid arthritis; were

obtained through a SNOMED-CT code record in GDPPR or an ICD-10 code in HES-APC for England, and through a Read V2 code in WLGP or an ICD-10 code in PEDW for Wales. Use of antidiabetic, antipsychotic medication, and cardiovascular disease prevention medication (including anticoagulant, antihypertensive, antiplatelet and statins) during the year before the index date was identified using BNF (British National Formulary) codes in Primary Care Meds for England, and in WDDS for Wales.

The code lists for all phenotypes are available on GitHub (https://github.com/BHFDSC/CCU037_02).

**Patient and public involvement (PPI)**

Four PPI representatives were included in the study team. In addition, three online meetings with a larger stakeholder group comprising patients, carers and members of the public from ethnically diverse backgrounds were conducted to get input into the study design, review initial results and to discuss how best to disseminate the study results to the public, leading to the design of a poster and infographic to share the results and encourage individuals to "Be proud of your ethnicity"[50].

In addition, this project was reviewed and approved by the UK National Institute for Health Research-British Heart Foundation (BHF) Cardiovascular Partnership lay panel that is comprised by individuals affected by cardiovascular disease.

**Statistical analysis**

The prevalence of demographic and clinical characteristics of the selected individuals was reported at the time of COVID-19 diagnosis, and was stratified by the following ethnicity classifications: the high-level ethnic groups, the 10 ethnic codes for Welsh patients, and the 19 NHS ethnicity codes for English patients.

All n (%) values extracted from the NHS England SDE and the SAIL Databank were rounded to the nearest multiple of 5, and counts < 10 were masked to minimise disclosure risk. The sum of counts may not return the exact value due to this rule.

**Age-standardised incidence rates.** All age-standardised incidence rates were stratified by sex and ethnicity. To estimate the age-standardised incidence rates (IR) for each study outcome and stratum, the age-specific IR were calculated for 5-year age bands (i.e., 30 to 34 years, 35 to 39 years, etc.) and then combined into age-specific estimates using the 2013 European Standard Population weights from 30 to 90+ age groups.

Estimates are reported per 100,000 population. Where there where ≥100 cases values are reported with 95% confidence intervals (CI), based on normal distribution; when cases were <100 and ≥10, Byar's approximation of the exact Poisson distribution was used, and where <10 cases were found, the exact Poisson approximation was used to quantify the confidence of the estimate.

In addition, to observe how the estimates varied across the different phases of the pandemic[10,51], we conducted a sub-analysis where individuals were stratified by the time of recorded COVID-19 diagnosis into their respective 6-monthly windows (see section Study design and duration for further details).

**Survival analysis.** Risk estimates for both outcomes of the study stratified by sex and ethnicity were calculated using a multivariable Cox regression model. The White or White British ethnic groups were used as the reference group. In the stratification by time, the reference group consisted of the White or White British populations from the corresponding time window. Reported models were adjusted by age, second order interaction of age, deprivation index (i.e., IMD), pregnancy status (only in women strata), prior COVID-19 vaccination, geographic location (only for England), period of COVID-19 recorded diagnosis, comorbidities and medication, location, and prior comorbidities recorded at the moment of COVID-19 diagnosis. Adjustment variables did not contain missing data. Hazard ratios (HR) are reported with 95% confidence intervals, and White British individuals were used as a reference group.

The list of covariates included in the adjustment was obtained from a list of key risk factors for addressing confounding in COVID-19 mortality, provided by the ONS[34], along with a list of cardiovascular confounders obtained from a predictive tool for cardiovascular risk developed and validated in the UK[52].

### Reporting summary

Further information on research design is available in the Nature Portfolio Reporting Summary linked to this article.

## Code availability

In the NHS England, SDE data were prepared using Python V.3.7 and Spark SQL (V.2.4.5) on Databricks Runtime V.6.4 for Machine Learning. Data were analysed using Python in Databricks and RStudio (Professional) Version 1.3.1093.1, driven by R Version 4.0.3. In SAIL, data were prepared and analysed using SQL and RStudio (Professional) Version 1.3.1093.1, driven by R Version 4.0.3. All code for data preparation and analysis are available on GitHub (https://github.com/BHFDSC/CCU037_02).

## Data availability

The data used in this study are available in NHS England's Secure Data Environment (SDE) service for England, but, as restrictions apply, they are not publicly available (https://digital.nhs.uk/services/secure-data-environment-service). The CVD-COVID-UK/COVID-IMPACT programme led by the BHF Data Science Centre (https://bhfdatasciencecentre.org/) received approval to access data in NHS England's SDE service for England from the Independent Group Advising on the Release of Data (IGARD) (https://digital.nhs.uk/about-nhs-digital/corporate-information-and-documents/independent-group-advising-on-the-release-of-data) via an application made in the Data Access Request Service (DARS) Online system (ref. DARS-NIC-381078-Y9C5K) (https://digital.nhs.uk/services/data-access-request-service-dars/dars-products-and-services). The CVD-

COVID-UK/COVID-IMPACT Approvals & Oversight Board (https://bhfdatasciencecentre.org/areas/cvd-covid-uk-covid-impact/) subsequently granted approval to this project to access the data within NHS England's SDE service for England and the Secure Anonymised Information Linkage (SAIL) Databank. The de-identified data used in this study were made available to accredited researchers only. Those wishing to gain access to the data should contact bhfdsc@hdruk.ac.uk in the first instance.

The data used in this study are available in the SAIL Databank at Swansea University, Swansea, UK, but as restrictions apply, they are not publicly available. All proposals to use SAIL data are subject to review by an independent Information Governance Review Panel (IGRP). Before any data can be accessed, approval must be given by the IGRP. The IGRP gives careful consideration to each project to ensure proper and appropriate use of SAIL data. When access has been granted, it is gained through a privacy protecting safe haven and remote access system referred to as the SAIL Gateway. SAIL has established an application process to be followed by anyone who would like to access data via SAIL at https://www.saildatabank.com/application-process.

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

## Acknowledgements

The British Heart Foundation Data Science Centre (grant No SP/19/3/34678, awarded to Health Data Research (HDR) UK), funded co-development (with NHS England) of the Secure Data Environment service for England, provision of linked datasets, data access, user software licences, computational usage, and data management and wrangling support, with additional contributions from the HDR UK Data and Connectivity component of the UK Government Chief Scientific Adviser's National Core Studies programme to coordinate national COVID-19 priority research. Consortium partner organisations funded the time of contributing data analysts, biostatisticians, epidemiologists, and clinicians. This work was carried out with the support of the BHF Data Science Centre led by HDR UK (BHF Grant no. SP/19/3/34678). This study made use of de-identified data held in NHS England's Secure Data Environment service for England and made available via the BHF Data Science Centre's CVD-COVID-UK/COVID-IMPACT consortium. This work used data provided by patients and collected by the NHS as part of their care and support. We would like to acknowledge all data providers who make health-relevant data available for research. This research is part of the Data and Connectivity National Core Study, led by Health Data Research UK in partnership with the Office for National Statistics and funded by UK Research and Innovation (grant ref MC_PC_20058). This work was also supported by The Alan Turing Institute via 'Towards Turing 2.0' EPSRC Grant Funding. The research was supported by the National Institute for Health and Care Research (NIHR) Oxford Biomedical Research Centre (BRC). DPA is funded through an NIHR Senior Research Fellowship (Grant number SRF-2018-11-ST2-004). The views expressed in this publication are those of the author(s) and not necessarily those of NHS England, the National Institute for Health and Care Research or the Department of Health. This study makes use of anonymised data held in the Secure Anonymised Information Linkage (SAIL) Databank. This work uses data provided by patients and collected by the NHS as part of their care and support. We would also like to acknowledge all data providers who make anonymised data available for research. We wish to acknowledge the collaborative partnership that enabled acquisition and access to the de-identified data, which led to this output. The collaboration was led by the Swansea University Health Data Research UK team under the direction of the Welsh Government Technical Advisory Cell (TAC) and includes the following groups and organisations: the SAIL Databank, Administrative Data Research (ADR) Wales, Digital Health and Care Wales (DHCW), Public Health Wales, NHS Shared Services Partnership (NWSSP) and the Welsh Ambulance Service Trust (WAST). All research conducted has been completed under the permission and approval of the SAIL independent Information Governance Review Panel (IGRP) project number 0911. This work was supported by the Con-COV team funded by the Medical Research Council (grant number: MR/V028367/1). This work was supported by Health Data Research UK, which receives its funding from HDR UK Ltd (HDR-9006) funded by the UK Medical Research Council, Engineering and Physical Sciences Research Council, Economic and Social Research Council, Department of Health and Social Care (England), Chief Scientist Office of the Scottish Government Health and Social Care Directorates, Health and Social Care Research and Development Division (Welsh Government), Public Health Agency (Northern Ireland), British Heart Foundation (BHF) and the Wellcome Trust. This work was supported by the ADR Wales programme of work. The ADR Wales programme of work is aligned to the priority themes as identified in the Welsh Government's national strategy: Prosperity for All. ADR Wales brings together data science experts at Swansea University Medical School, staff from the Wales Institute of Social and Economic Research, Data and Methods (WISERD) at Cardiff University and specialist teams within the Welsh Government to develop new evidence which supports Prosperity for All by using the SAIL Databank at Swansea University, to link and analyse anonymised data. ADR Wales is part of the Economic and Social Research Council (part of UK Research and Innovation) funded ADR UK (grant ES/S007393/1). This work was supported by the Wales COVID-19 Evidence Centre, funded by Health and Care Research Wales.

## Author contributions

Conceptualisation: S.K., D.P.A., A.D. and G.C. Data curation for England: M.P.M. and F.A. Formal analysis for England: M.P.M. and F.A. Data curation for Wales: D.P. and H.A. Formal analysis for Wales: H.A. Funding acquisition: S.K. Data interpretation: M.P.M. and S.K. Writing original draft: M.P.M., S.K. Writing review and editing: all authors. Approving the final version of the manuscript: all authors. S.K. and M.P.M. take responsibility for the integrity of the data analysis.

## Competing interests

K.K. is a chair of the Ethnicity Subgroup of the UK Scientific Advisory Group for Emergences (SAGE), and a member of SAGE. This work was also supported by The Alan Turing Institute via 'Towards Turing 2.0' EPSRC Grant Funding. DPA's research group has received grant/s from Amgen, Chiesi-Taylor, Lilly, Janssen, Novartis, and UCB Biopharma, and consultancy fees from AstraZeneca and UCB Biopharma. Amgen, Astellas, Janssen, Synapse Management Partners and UCB Biopharma have funded or supported training programmes organised by S.K. and DPA's department. S.K. receives funding support from Amgen BioPharma outside of this work. This research is part of the Data and Connectivity National Core Study, led by Health Data Research UK in partnership with the Office for National Statistics and funded by UK Research and Innovation (grant ref MC_PC_20058). The remaining authors have nothing to declare.

## Additional information

[1]Centre for Statistics in Medicine, Nuffield Department of Orthopaedics, Rheumatology and Musculoskeletal Sciences (NDORMS), University of Oxford, Oxford, UK. [2]Institute of Health Informatics, 222 Euston Road, London, NW1 2DA. University College London, London, UK. [3]UK Research and Innovation Centre for Doctoral Training in AI-enabled Healthcare Systems, University College London Hospitals Biomedical Research Centre, University College London, London, UK. [4]Population Data Science, Swansea University Medical School, Faculty of Medicine, Health & Life Science, Swansea University, Swansea, UK. [5]British Heart Foundation Cardiovascular Epidemiology Unit, Department of Public Health and Primary Care, and Cambridge Centre for AI in Medicine, University of Cambridge, Cambridge, UK. [6]British Heart Foundation Data Science Centre, Health Data Research UK, London, UK. [7]Diabetes Research Centre, Leicester Diabetes Research Centre, University of Leicester, Leicester, UK. [8]Department of Orthopaedic Surgery & Sports Medicine, Erasmus MC University Medical Centre Rotterdam, Rotterdam, The Netherlands. ✉e-mail: sara.khalid@ndorms.ox.ac.uk

## the CVD-COVID-UK/COVID-IMPACT Consortium

Marta Pineda-Moncusí ◉[1], Freya Allery ◉[2,3], Hoda Abbasizanjani ◉[4], David Powell[4], Albert Prats-Uribe ◉[1], Johan H. Thygesen ◉[2], Angela Wood ◉[5,6], Christopher Tomlinson ◉[2], Amitava Banerjee ◉[2], Ashley Akbari ◉[4], Antonella Delmestri ◉[1], Laura C Coates ◉[1], Spiros Denaxas[2,6], Kamlesh Khunti[7], Gary Collins ◉[1], Daniel Prieto-Alhambra ◉[1,8] & Sara Khalid ◉[1]✉

