## [Transparent Peer Review file · Nature Communications]

Ethnic disparities in COVID-19 mortality and cardiovascular disease in England and Wales between 2020-2022

Corresponding Author: Professor Sara Khalid

Version 0:

Reviewer comments:

Reviewer #1

(Remarks to the Author)

This paper studies rich, individual-level data from England and Wales covering the period 23 January 2020 to 29 June 2022. Detailed results on disparities in outcomes by ethnicity are presented. The authors had a rare opportunity to consider very valuable data. This seems a nice piece of descriptive epidemiology. While the results are very rich, there is no analysis that tells us what should be done in response to these disparities.

The authors study individuals with a diagnosis of COVID-19 recorded in their primary care record or when admitted to hospital. This produces a cohort of 4.9M. The index date for the analysis is the date of COVID-19 diagnosis, but it is not stated if this is only the first or any COVID-19 diagnosis for a specific individual. Regardless, the population of 4.9M must represent a small subset of COVID-19 infections. The paragraph starting on line 441 discusses this problem of relying on a small subset of COVID-19 infections, but the potential biases are not discussed in detail and the real possibility that some of the presented results are substantially affected by this remains.

The authors examine two outcomes, but do not tell us why. The focus on incidence of CVD, for example, is mentioned very briefly only in the Discussion section.

There are no data on the two outcomes from before the COVID-19 pandemic, so it is not clear if these are new disparities or continuations of previous disparities. That does not lessen their importance but makes a very large difference to what we should learn from them.

How the six-month periods within the overall study period are considered is not made very clear. Is there a reference period or is each ethnic group compared to a common reference group in each period separately? If the latter, the results are critically based on what happens to outcomes in the reference category over time. This is not discussed in the manuscript.

Specific comments

Abstract

- Doesn't start with a statement of what we know and what we don't know and therefore the knowledge gap that this paper fills is not clear
- Focuses on CVD incidence and mortality but does not explain why
- Some of the statements of results in the second paragraph are not clear whether they are describing differences over time or are relative to a reference category (or both perhaps)

Lines 134-137: I did not understand the differences between the two sources of data on hospital admissions in England

Line 139: the reason for the age restriction is not justified. Does this impact some ethnicities more than others?

Line 146: I think this is any-cause mortality, but it is not stated clearly

Line 148: the reason for focusing on cardiovascular disease is not stated here, or in the introduction even. The first point at which this is discussed is line 412

Line 227: detailed of how models were adjusted for “deprivation index” are not given

Line 229: is “location” the same as “location” on line 228?

Line 234: the cohort of 4.9M individuals with a diagnosis of COVID-19 seems very low

Line 237-241: how do these ethnicity compositions compare with other data sources?

(Remarks on code availability)

Reviewer #2

(Remarks to the Author)

Thank you to the authors for this interesting paper examining ethnic and racial disparities in severe COVID-19 outcomes in England and Wales. This study makes an important contribution by using comprehensive data and granular ethnicity categories to document the continued inequities in COVID-19 outcomes, specifically 28-day mortality and 30-day cardiovascular disease acute event (CVD). They leverage large national datasets to highlight inequities occurring within the larger racial and ethnic categories that are typically utilized in research. The paper has potential, however there are a couple areas of concern that should be addressed: 1) aligning the methods, results, tables, and figures; and 2) the definition, discussion, and interpretation of race, ethnicity and disparities.

Aligning the methods, results, tables, and figures:

It is hard to follow the comparisons between the different classification systems in the figures. If comparing the results by classification system is a central research question, then it would be helpful to create a figure or table that more clearly shows the differences between the results from different classifications systems.

The data are reported by sex, but no conclusions are discussed in the text. If sex differences represent an important dimension of rate differences, then these results should be reported. If not, then the results may not need to be presented by sex.

The hazard models do not seem like they add much to the paper - they are not discussed in the conclusion and do not seem to show substantively different results from the main models. They also introduce new sources of bias (e.g., time to diagnosis). The authors should consider cutting these results-- this will also help to streamline the results section so it's more concise.

Discussion of race and ethnicity:

One major point is how race and ethnicity are discussed in the article. The authors should be explicit about 1) how they are defining and conceptualizing race and ethnicity and 2) the interpretations and implications of racial and ethnic health disparities. It is great that the authors explicitly state how they categorized each racial and ethnic category (according to the four different classification schemas). The difference between race and ethnicity is not articulated and there is no indication about how race and ethnicity were reported. Were they self-reported/self-defined by the patient? Assigned by the physician? Even more importantly, there is no discussion about how the authors are interpreting racial and/or ethnic disparities in COVID-19 severe outcomes. Are the authors implying that there are underlying biological differences between the race and ethnic groups that make them more likely to have severe COVID-19 outcomes? Or are they implying that there are underlying social and structural causes including racism and classism that produce inequitable conditions and unique vulnerabilities to COVID-19 severe outcomes? Please clarify. Nature offers some guidance on this topic: <https://www.nature.com/articles/d41586-023-00973-7>

Additional edits

Occupation (e.g. essential worker vs not) is another potential important sociodemographic characteristic to consider, given substantial occupational segregation by race-ethnicity. If available, the authors should consider including this variable.

In the discussion, the authors write that the mortality gap disappeared after the appearance of omicron. Further theorizing about this change would be helpful-for example, do the authors believe this change is related to the timing of vaccine introduction, or something to do with the omicron variant?

The CVD outcome should be introduced in the introduction, and then further detail on why it is included, and its

conceptualization should be added to the methods. For example, how did the authors identify a new CVD diagnosis versus a prior diagnosis?

The authors include a number of covariates in the analysis. It would help to conceptualize why these are included—are they confounders? Minor revisions:

It is not clear what the percentages in the second paragraph of the abstract mean – please specify

The paper states that in the survival analysis the “adjustment variables did not contain missing data” does that mean that a complete case analysis (i.e. excluded those with missing data) was conducted or that the variables had no missing data? Please specify.

Using line graphs for the figures that show data over time would make it easier to see the trends.

A clear statement about the gap this article is addressing would be helpful.\

Did the authors use direct or indirect age adjustment? Related, some studies in the US have found even wider racial-ethnic disparities in middle ages (e.g. 40-50)- the authors could consider a sensitivity analysis using age bands rather than age adjustment.

What is the geography for the IMD? Neighborhood?

It would be helpful to include a sensitivity analysis including those with less than 1 year of primary care records. This can help address potential selection bias around access to primary care.

Would suggest that the authors list the various data sources in a table rather than as a bulleted list.

Figures five and six are quite hard to read because the text is so small and there's so much information in each panel.

The figures after Figure 7 do not have titles. Are they supposed to be part of the supplementary materials? If so, it would still be useful to include the titles. It would also be helpful to see a table of contents for the supplementary materials.

(Remarks on code availability)

Reviewer #3

(Remarks to the Author)

(Remarks on code availability)

Reviewer #4

(Remarks to the Author)

This manuscript “Are ethnic disparities in COVID-19 severe outcomes over? Analysis of 5.3 million individuals in England and Wales from 2020-2022” presents from an analysis of linked electronic health data among adults by ethnic category to compare 30 day risk of mortality and cardiovascular disease. The COVID-19 pandemic highlighted the importance of understanding health disparities with the recognition individuals with high social vulnerability or minority ethnicities were more likely to be infected with SARS-CoV-2 and have severe outcomes from acute COVID-19. These disparities were recognized early in 2020 but it is unclear whether these disparities have persisted. Further, many studies in this area have used broad ethnic categories, limiting the ability to develop more targeted interventions or to understand potential factors contributing to the disparities. These are two reasons why this study provides important information and contributes to our understanding of severe COVID-19 outcomes.

One of the main objectives of this manuscript, as relayed by the title, is to examine how disparities by the outcomes and by ethnic groups may have changed over time. There is extensive literature in this area as to how to measure health disparities, how to compare differences across groups and over time. This manuscript would be strengthened by a review of this literature and re-analyses. For example, figure 5 that reports the incidence by ethnicity group across the time periods, could be presented as trend lines for the incidence of mortality (e.g.) within each ethnic group. The slope and trajectory of each ethnic group line could be compared over time. This would provide information as to whether the incidence of mortality (e.g.) was decreasing more rapidly in one ethnic group compared to another. The authors could also compare the absolute difference in incidence for each group compared to Whites at time period 1 and then compare the absolute difference compared to Whites at the last time period. This would also provide information as to whether or not the gap between mortality by ethnic group is narrowing over time.

Detailed comments for the authors-

1. The title mentions "COVID-19 severe outcomes" but the authors do not include risk of hospitalization for acute COVID-19 or risk of ICU admission as severe outcomes. It is not clear why these are not included.
2. The authors define severe outcomes as mortality within the first 28 days or cardiovascular disease within the first 30 days after acute COVID-19. There are a number of questions related to the selection of these outcomes. First, it is not clear why the time frame for the two outcomes differs or why 28 days and 30 days were selected. Next, it is not clear why cardiovascular disease (CVD) is included in this paper.
3. The outcome of CVD could be a separate paper. This would also help focus and streamline the results.
4. Abstract (lines 41-42) and repeated in the conclusions- "COVID-19 mortality and CVD risk was increased in most non-White ethnic groups in England and Asian population in Wales during 2.5 years after the pandemic outbreak." This sentence could be misinterpreted to imply that the authors focused on outcomes 2.5 years after acute COVID-19. Recommend revising to more accurately report the results"...30-day mortality and 28 day CVD risk was higher among non-White ethnic groups..."
5. Reporting both the NHS and the SNOMED-CT ethnic groups is a valuable contribution of this analysis, but there are a lot of results. Recommend streamlining the paper by focusing only on the NHS ethnic groups, in addition to the 6-census groups in England and 10-level classification for Wales. This would reduce the number of panels for each figure. The authors can focus the results and discussion around these groupings. The results from the SNOMED-CT groups can be presented in the supplemental tables and then briefly discussed as to whether or not they add to this discussion on disparities.
6. Severity of acute COVID-19 (hospitalized, ICU, etc) and acute COVID-19 treatment are associated with increased risk of 30 day mortality. These clinical considerations need to be compared across ethnic groups and accounted for in adjusted analysis.
7. Footnote in the figure states that vaccination status is included in the adjusted variables. How was vaccination defined? Was this based on evidence of any vaccination? Vaccination at least 14 days prior to COVID-19? Then in the survival analysis section of the methods, vaccination status is not included in the list of adjusted variables. Please confirm whether or not vaccination status was included in the models.
8. Results: Did you observe differences in the demographic (age, sex) or clinical (co-morbidities, acute COVID-19 severity) across ethnic groups over time? In other words, did the case mix of COVID-19 patients differ over time by ethnic group? Were there differences over time in England compared to Wales?
9. Line 263-266 could be moved to the methods
10. There are strong reasons to stratify all the results by male, female. This reasoning needs to be added to both the introduction and the discussion.
11. Line 398, the first sentence should clarify that the ethnic groups had increased mortality compared to Whites. Please review entire manuscript to make sure it is clear who the comparison group is.
12. The discussion would be strengthened by including more details about ethnic disparities in general in England and Wales, whether or not they were present prior to COVID pandemic, how the pandemic did or did not exacerbate them, and potential reasons why they exist.
13. Line 469 states that reasons for these disparities are complex and intersectional, but these concepts and background are not presented in this manuscript.
14. Figures reporting incidence- add a footnote with details on the method and age groups for age standardization.
15. Figure 1 (e.g.) consider reporting only high level and NHS ethnic groups for England. This would allow the figure to be reduced to a and b- one for England and one for Wales, with 2 panels each -one for mortality and one for CVD.
16. Figure 2 (e.g.- applies to all HR figures) HR needs to start at 1.0 and then graph the estimate below or above 1.0. Consider also reporting the results on a log scale.

(Remarks on code availability)

Version 1:

Reviewer comments:

Reviewer #1

(Remarks to the Author)

I have focused this review on the responses to my initial comments.

In response to my first comment expressing concern about the descriptive nature of the research, the authors say: "To address such disparities, those responsible for delivering health care and regulators need to adapt the guidelines and health policies, only then an analysis to confirm that such guidelines and policy changes had addressed the disparities could be performed." I find this too vague to be useful and am concerned that the impact of the work will be limited.

My comment #2. I am confused by the authors' response. They state that there were over 21 million cases in the UK by 1st April 2022. They then say the number of individuals who had ever had COVID-19 was estimated to be 212,000 in Wales and 4.1 million in England by the same date. Are they saying that *on average* a person having COVID-19 at least once had COVID-19 five times within two years? That would certainly be worthy of comment if true.

My comment #4: The authors have clarified what they did and who told them to do that, but have not addressed this part of my comment "If the latter, the results are critically based on what happens to outcomes in the reference category over time. This is not discussed in the manuscript." If the rates in the reference category are changing over time, this is important context for the changes over time in the relative differences.

Responses to specific comments

These are generally fine, but I remain unclear on these:

My comment: "Abstract - Doesn't start with a statement of what we know and what we don't know and therefore the knowledge gap that this paper fills is not clear". Everyone faces the same word constraints. I don't think there is a need for additional space.

My comment: "Line 227: detailed of how models were adjusted for "deprivation index" are not given". The authors have told me WHAT deprivation measure they used, but not HOW they have adjusted for it. It is a continuous but not cardinal variable

Minor point: there are several typos in the responses and revised manuscript text

(Remarks on code availability)

Reviewer #2

(Remarks to the Author)

Thank you for addressing our comments and questions. The clarity of the manuscript is much improved. We have a couple remaining comments below.

2) The data are reported by sex, but no conclusions are discussed in the text. If sex differences represent an important dimension of rate differences, then these results should be reported. If not, then the results may not need to be presented by sex.

a. Follow up comment: I appreciate the additional context and justification added by the authors. It would also be valuable to highlight that while overall men have had higher rates over covid 19 mortality that is not true in all geographies or subgroups. It is useful to highlight the places or subgroups where women had higher rates (if applicable). Where you say "men generally had higher incidence..." you could also list the subgroups where women had higher incidence.

b. See the following citation: Danielsen AC, Lee KM, Boulicault M, Rushovich T, Gompers A, Tarrant A, Reiches M, Shattuck-Heidorn H, Miratrix LW, Richardson SS. Sex disparities in COVID-19 outcomes in the United States: Quantifying and contextualizing variation. *Soc Sci Med.* 2022 Feb;294:114716. doi: 10.1016/j.socscimed.2022.114716. Epub 2022 Jan 10. PMID: 35042136; PMCID: PMC8743486.

5.1) What is the geography for the IMD? Neighborhood?

a. Follow up comment: . Thank you for providing additional information about the IMD. In the statement that you added ("The IMD is obtained using a broad spectrum of indicators including income, barriers to housing and services, education and skills, employment, health and disability, crime, and living environment; and it is often categorised into fifths, with 1 denoting the most deprived and 5 the least deprived areas"), how is the area referenced defined? Is it neighborhoods, or another administrative or political geographic unit?

(Remarks on code availability)

Reviewer #3

(Remarks to the Author)

(Remarks on code availability)

Reviewer #4

(Remarks to the Author)

The authors have carefully addressed each reviewer's comments.

While the gap that this research fills is clearer now, and the focus on cardiovascular disease is clearer, the title of the paper should be revised. The refers to severe outcomes, but the analysis is focused on mortality and CVD. I recommend revising the title of the paper to reflect this.

(Remarks on code availability)

REVIEWER COMMENTS

Reviewer #1 (Remarks to the Author):

We would like to express our gratitude to the reviewer for the time and effort spend in helping us improving the quality of our manuscript.

1) This paper studies rich, individual-level data from England and Wales covering the period 23 January 2020 to 29 June 2022. Detailed results on disparities in outcomes by ethnicity are presented. The authors had a rare opportunity to consider very valuable data. This seems a nice piece of descriptive epidemiology. While the results are very rich, there is no analysis that tells us what should be done in response to these disparities.

We thank the reviewer for sharing the concern about what can be done to reduce the health disparity observed in the results. Our paper focuses on highlighting the inequities in mortality and CVD after being infected by SARS-Cov-2 as a first step to tackle this inequity. In line with prior findings but providing more granularity in the ethnic groups affected, we observed an increased COVID-19 mortality in non-White British population over the COVID-19 pandemic that closed after the entrance of omicron. Detected for the first time, we reported an increased risk of CVD in patients with SARS-Cov-2 among non-White British population over the pandemic. However, and in contrast to COVID-19 mortality, the disparity on CVD did not close by the end of the study. To address such disparities, those responsible for delivering health care and regulators need to adapt the guidelines and health policies, only then an analysis to confirm that such guidelines and policy changes had addressed the disparities could be performed. The analysis we undertook actually shows where the gaps are and where healthcare providers can focus.

Following the reviewer's suggestion, we had included recommendations to tackle these disparities in the discussion.

Key revisions now included:

"The analysis we undertook shows where the gaps on COVID-19 mortality and CVD are and where healthcare providers can focus to address them."

2) The authors study individuals with a diagnosis of COVID-19 recorded in their primary care record or when admitted to hospital. This produces a cohort of 4.9M. The index date for the analysis is the date of COVID-19 diagnosis, but it is not stated if this is only the first or any COVID-19 diagnosis for a specific individual. Regardless, the population of 4.9M must represent a small subset of COVID-19 infections. The paragraph starting on line 441 discusses this problem of relying on a small subset of COVID-19 infections, but the potential biases are not discussed in detail and the real possibility that some of the presented results are substantially affected by this remains.

Thank you for this observation. According to the Johns Hopkins statistics on COVID-19 (<https://coronavirus.jhu.edu/region/united-kingdom>), there were 21.4M of cumulative cases of SARS-Cov-2 infections in the UK from its onset to 1st of April 2022. Our analysis focuses only in fist COVID-19 diagnosis during the same time period, as described under the participants subsection: we are capturing participants with a first record of confirmed COVID-19 diagnosis between 23rd January 2020 and 1st April 2022 among individuals between 30 and 100 years, and that index date was the first recorded date of COVID-19 diagnosis. The

statistics between 2020 and 2022 of new cases of COVID-19 in the UK, which were provided by the UK Health Security Agency (UKHSA) are not available anymore. When we selected the number of patients from the database after applying our inclusion criteria, the number of patients was consistent with the total number of values reported by the UKHSA at the time. We could find a report from the Welsh government* dated in 1st of April 2022, which estimated that number of individuals who had COVID at least once in Wales oscillated between 212,000 people (95%CI: 189,800 to 234,800) and in England between 4,122,700 people (95CI%: 4,013,600 to 4,228,300), which is lower than the number of patients included in this study. Thus, we consider our sample size representative of the COVID-19 population for that period. At the same time, we fully acknowledge the reviewer's point about potential biases due to sample size, and would like to clarify that whilst these numbers may not include the whole general population, are they representative of the actual cases in the real world healthcare settings in England and Wales.

*Table 1 from <https://www.gov.wales/coronavirus-covid-19-infection-survey-positivity-estimates-20-26-march-2022-html>

Line 441 refers to individuals who might have had COVID-19 but never reported and therefore were never registered in the electronic health records, and that won't be reported either by official organisms like the ONS.

Key revisions now included: we have now incorporated the above details and clarifications in the main text, within Strengths and limitations of this study:

“Despite the cohorts of the study might not include the whole new cases of COVID-19 in the UK during the study period, the data sources of the study have been previously considered representative of the England and Welsh populations.”

3) The authors examine two outcomes, but do not tell us why. The focus on incidence of CVD, for example, is mentioned very briefly only in the Discussion section. There are no data on the two outcomes from before the COVID-19 pandemic, so it is not clear if these are new disparities or continuations of previous disparities. That does not lessen their importance but makes a very large difference to what we should learn from them.

We thank the reviewer for the suggestion. We decided to include CVD as an outcome of interest for COVID-19 given that cardiac injury is recognized as one of the most frequent complications of the disease. Furthermore, despite it is well known that certain ethnic groups such as Black have higher risk of CVD, we identified there was a gap in the literature where risk of cardiovascular events after SARS-CoV-2 infections across different ethnic groups was not explored.

Following the reviewer's advice, we have expanded the rational to include CVD in the introduction and at the beginning off the discussion.

Key revisions now included:

Introduction:

“On the other hand, prior studies have associated COVID-19 with many cardiac complications,¹⁰ and an increased mortality risk in people with history of cardiovascular disease (CVD).¹¹ The ONS had reported a higher CVD mortality in ethnic minorities,¹² however, little is known about the differences CVD across ethnic groups after COVID-19 infection.¹³”

10. Basu-Ray I, Almaddah Nk, Adeboye A, et al. Cardiac Manifestations of Coronavirus (COVID-19) [Updated 2024 Feb 12]. In: StatPearls [Internet]. Treasure Island (FL): StatPearls Publishing; 2024 Jan-. Available from: <https://www.ncbi.nlm.nih.gov/books/NBK556152>

11. Singh, A. K. et al. Prevalence of co-morbidities and their association with mortality in patients with COVID-19: A systematic review and meta-analysis. *Diabetes Obes Metab* 22, 1915-1924, doi:10.1111/dom.14124 (2020).

12. Ali, R., Raleigh, V., Majeed, A. & Khunti, K. Life expectancy by ethnic group in England. *BMJ* 375, e068537, doi:10.1136/bmj-2021-068537 (2021).

13. Islam, S. J. et al. County-Level Social Vulnerability is Associated With In-Hospital Death and Major Adverse Cardiovascular Events in Patients Hospitalized With COVID-19: An Analysis of the American Heart Association COVID-19 Cardiovascular Disease Registry. *Circ Cardiovasc Qual Outcomes* 15, e008612, doi:10.1161/CIRCOUTCOMES.121.008612 (2022).

Discussion:

“Prior studies reported men had higher risk of COVID-19 mortality when compared to women.^{3-5,32} Additionally, CVD is more prevalent in men than women in the general population.³³ Thus, we analysed the risk of COVID-19 mortality and CVD separately for men and women. Our results confirm men generally had higher incidence and risk of COVID-19 mortality than women. Likewise, men had higher incidence and risk of CVD, except for Middle Eastern women, whose incidence of CVD was higher than Middle Eastern man.”

Additionally, we added in the Strengths and limitations section the following clarification of the importance of having CVD as an outcome for COVID-19 research:

“Our study fills a gap in the literature regarding the risk of CVD after SARS-Cov-2 infection across different ethnic groups.”

3. Ramirez-Soto, M. C., Ortega-Caceres, G. & Arroyo-Hernandez, H. Sex differences in COVID-19 fatality rate and risk of death: An analysis in 73 countries, 2020-2021. *Infez Med* 29, 402-407, doi:10.53854/liim-2903-11 (2021).

4. Doerre, A. & Doblhammer, G. The influence of gender on COVID-19 infections and mortality in Germany: Insights from age- and gender-specific modeling of contact rates, infections, and deaths in the early phase of the pandemic. *PLoS One* 17, e0268119, doi:10.1371/journal.pone.0268119 (2022).

5. Fabiao, J. et al. Why do men have worse COVID-19-related outcomes? A systematic review and meta-analysis with sex adjusted for age. *Braz J Med Biol Res* 55, e11711, doi:10.1590/1414-431X2021e11711 (2022).

32. Office-for-National-Statistics. Coronavirus (COVID-19) and the different effects on men and women in the UK, March 2020 to February 2021, <<https://www.ons.gov.uk/peoplepopulationandcommunity/healthandsocialcare/conditionsanddiseases/articles/coronaviruscovid19andthedifferenteffectsonmenandwomenintheukmarch2020tofebruary2021/2021-03-10>> (2021).

33. Bots, S. H., Peters, S. A. E. & Woodward, M. Sex differences in coronary heart disease and stroke mortality: a global assessment of the effect of ageing between 1980 and 2010. *BMJ Global Health* 2, e000298, doi:10.1136/bmjgh-2017-000298 (2017).

4) How the six-month periods within the overall study period are considered is not made very clear. Is there a reference period or is each ethnic group compared to a common reference group in each period separately? If the latter, the results are critically based on what happens to outcomes in the reference category over time. This is not discussed in the manuscript.

We thank the reviewer for pointing this out. This study design was recommended through consensus by the CVD-COVID-UK / COVID-IMPACT consortium, made due to the complexity of studying the entire pandemic period rigorously and given the temporal variability in terms of COVID strains, available testing, seasonality, and lockdown rules. Thus, it was decided a priori (at the analysis plan step) to split the 2.5 years in 6-month periods as a programmatically decision to explore study period over time. Each window

shows the IR of that 6- months, and in case of the HR, the reference group is the White/White British population of that same 6-month period. We have clarified which group was used as the reference in the Statistical methods (within the survival analysis subsection), as well as in the figures' legend.

Key revisions now included:

Methods - Statistical analysis: Survival analysis subsection:

“White or White British ethnic groups were used as the reference group. In the stratification by time, the reference group was the White or White British populations from that particular time window.”

5) Specific comments

Abstract

- Doesn't start with a statement of what we know and what we don't know and therefore the knowledge gap that this paper fills is not clear

We appreciate the reviewer's suggestion. However, the maximum number of words allowed in the abstract prevent us to include more rationale as suggested. If the editor would allow us to exceed it, we are more than happy to include a statement regarding the gap that our paper fills. Which would be something in line with: “Our study fills a gap in the literature regarding the risk of CVD after SARS-Cov-2 infection across different ethnic groups”.

- Focuses on CVD incidence and mortality but does not explain why

Thank you. We have expanded the rationale as requested above.

- Some of the statements of results in the second paragraph are not clear whether they are describing differences over time or are relative to a reference category (or both perhaps)

We acknowledge and appreciate the reviewer's comment. We were unsure if the review referred to the second paragraph starting in line 243 from the original submitted document, which introduced the baseline characteristics tables of the patients. Baseline characteristics are not stratified over time, as specified in the statistical analysis details: Prevalence of demographic and clinical characteristics of the selected individuals are reported at the time of COVID-19 diagnosis. We have expanded this statement to include that the characteristics were stratified by ethnicity.

Key revisions now included:

“Prevalence of demographic and clinical characteristics of the selected individuals are reported at the time of COVID-19 diagnosis, and were stratified by the following ethnicity classifications: the high-level ethnic groups, the 10 ethnic codes for the Welsh patients, and the 19 NHS ethnicity codes for the English patients.”

Lines 134-137: I did not understand the differences between the two sources of data on hospital admissions in England

We appreciate the reviewer's concern in this topic. This is particular to how hospital data is organised and processed in England. The APC (admitted patient care) dataset is the national data set for hospital admissions and is obtained directly from the patient's clinical records. The other dataset, SUS (Secondary Uses Service), was built for purposes other than direct clinical management and thus complements and contains more information than APC, such as administrative information. We have clarified the difference in Table 1 legend.

Key revisions now included:

Table 1 legend.

¹ HES-APC is the national data set for hospital admissions and is obtained directly from the patient's clinical records.

² SUS was built for purposes other than direct clinical management and thus complements and contains more information than HES-APC, such as administrative information.”

Line 139: the reason for the age restriction is not justified. Does this impact some ethnicities more than others?

As specified in the Strengths and limitations section, we included the age restriction following the methodology developed by the ONS. The main reason they implemented such age restrictions, which we agreed was suitable for the scope of our research, is that the incidence of severe COVID-19 outcomes below the age of 30 years was very rare, whilst individuals older than 100 years were uncommon. To account for the age differences across ethnic groups, we include the age standardisation in the incidence analysis, and we adjusted by age in the survival analysis. We have included the rationale in the Strengths and limitations section.

Key revisions now included:

Strengths and limitations of this study

“To calculate our age-standardised IR, we replicated the ONS methodology, including the restriction of the population to ages from 30 to 100, but included 19 rather than 10 ethnic groups in the England population.²⁰ The age restriction was applied because the incidence of severe COVID-19 outcomes in individuals under 30 years of age was very rare, while those older than 100 years were uncommon.”

Line 146: I think this is any-cause mortality, but it is not stated clearly

Thank you for the clarification. The 28-day death measure was used as a near real-time proxy of COVID-19 deaths for a rapid reporting (on a daily basis) of the status of the pandemic in the GOV.UK COVID-19 Dashboard (<https://ukhsa-dashboard.data.gov.uk/>). [<https://doi.org/10.1093/ije/dyad116>]

Since we did not have the specific-cause of death, we used the 28-day proxy, which has been also used before in the literature for COVID-19 mortality. For instance by Lavrentieva, A., *et al.* (2023) or Thygesen, J. H. *et al.* (2025). We have specified that we included any death within the 28 days in the Methods - Outcomes of the study" section, to clarify it, and added the example references.

Key revisions now included:

Methods - Outcomes of the study

“28-day mortality: defined as any death within the 28 days from COVID-19 event (positive test, diagnosis, or hospital admission due to COVID-19).^{21,22}”

21. Lavrentieva, A., *et al.* “An observational study on factors associated with ICU mortality in Covid-19 patients and critical review of the literature.” *Sci Rep* **13**, 7804 (2023). <https://doi.org/10.1038/s41598-023-34613-x>

22. Thygesen, J. H. *et al.* “Prevalence and demographics of 331 rare diseases and associated COVID-19-related mortality among 58 million individuals: a nationwide

retrospective observational study.” *The Lancet. Digital health* vol. 7,2 (2025): e145-e156.
doi:10.1016/S2589-7500(24)00253-X

Line 148: the reason for focusing on cardiovascular disease is not stated here, or in the introduction even. The first point at which this is discussed is line 412

Thank you for bringing this matter to our attention. We have introduced the reason for focusing on CVD in the introduction and at the beginning of the discussion.

Key revisions now included:

Introduction:

“On the other hand, prior studies have associated COVID-19 with many cardiac complications,¹⁰ and an increased mortality risk in people with history of cardiovascular disease (CVD).¹¹ The ONS had reported a higher CVD mortality in ethnic minorities,¹² however, little is known about the differences CVD across ethnic groups after COVID-19 infection.¹³”

10. Basu-Ray I, Almaddah Nk, Adeboye A, et al. Cardiac Manifestations of Coronavirus (COVID-19) [Updated 2024 Feb 12]. In: StatPearls [Internet]. Treasure Island (FL): StatPearls Publishing; 2024 Jan-. Available from: <https://www.ncbi.nlm.nih.gov/books/NBK556152>

11. Singh, A. K. et al. Prevalence of co-morbidities and their association with mortality in patients with COVID-19: A systematic review and meta-analysis. *Diabetes Obes Metab* 22, 1915-1924, doi:10.1111/dom.14124 (2020).

12. Ali, R., Raleigh, V., Majeed, A. & Khunti, K. Life expectancy by ethnic group in England. *BMJ* 375, e068537, doi:10.1136/bmj-2021-068537 (2021).

13. Islam, S. J. et al. County-Level Social Vulnerability is Associated With In-Hospital Death and Major Adverse Cardiovascular Events in Patients Hospitalized With COVID-19: An Analysis of the American Heart Association COVID-19 Cardiovascular Disease Registry. *Circ Cardiovasc Qual Outcomes* 15, e008612, doi:10.1161/CIRCOUTCOMES.121.008612 (2022).

Discussion:

“We analysed their risk of mortality and CVD during the 2.5 years after the pandemic outbreak. Prior studies reported men had higher risk of COVID-19 mortality when compared to women.^{3-5,32} Additionally, CVD is more prevalent in men than women in the general population.³³ Thus, we analysed the risk of COVID-19 mortality and CVD separately for men and women. Our results confirm men generally had higher incidence and risk of COVID-19 mortality than women. Likewise, men had higher incidence and risk of CVD, except for Middle Eastern women, whose incidence of CVD was higher than Middle Eastern man.”

3. Ramirez-Soto, M. C., Ortega-Caceres, G. & Arroyo-Hernandez, H. Sex differences in COVID-19 fatality rate and risk of death: An analysis in 73 countries, 2020-2021. *Infez Med* 29, 402-407, doi:10.53854/liim-2903-11 (2021).

4. Doerre, A. & Doblhammer, G. The influence of gender on COVID-19 infections and mortality in Germany: Insights from age- and gender-specific modeling of contact rates, infections, and deaths in the early phase of the pandemic. *PLoS One* 17, e0268119, doi:10.1371/journal.pone.0268119 (2022).

5. Fabiao, J. et al. Why do men have worse COVID-19-related outcomes? A systematic review and meta-analysis with sex adjusted for age. *Braz J Med Biol Res* 55, e11711, doi:10.1590/1414-431X2021e11711 (2022).

32. Office-for-National-Statistics. Coronavirus (COVID-19) and the different effects on men and women in the UK, March 2020 to February 2021, <https://www.ons.gov.uk/peoplepopulationandcommunity/healthandsocialcare/conditionsanddiseases/articles/coronaviruscovid19andthedifferenteffectsonmenandwomenintheukmarch2020tofebruary2021/2021-03-10> (2021).

33. Bots, S. H., Peters, S. A. E. & Woodward, M. Sex differences in coronary heart disease and stroke mortality: a global assessment of the effect of ageing between 1980 and 2010. *BMJ Global Health* 2, e000298, doi:10.1136/bmjgh-2017-000298 (2017).

Line 227: detailed of how models were adjusted for “deprivation index” are not given

We thank the reviewer for the insightful inquiry. We used the multiple deprivation index, which is detailed in the covariate section. We have clarified in the Methods - Survival analysis section that when adjusting for multiple deprivation we used the IMD.

Key revisions now included:

“ Reported models were adjusted by age, second order interaction of age, deprivation index (i.e., IMD), pregnancy status (only in women strata), prior COVID-19 vaccination, geographic location (only for England), period of COVID-19 recorded diagnosis, comorbidities and medication, location, and prior comorbidities recorded at the moment of COVID-19 diagnosis”

Line 229: is “location” the same as “location” on line 228?

Thanks for spotting this. Yes, it was the same location. We deleted the second “location” in line 229.

Line 234: the cohort of 4.9M individuals with a diagnosis of COVID-19 seems very low

Thank you for the consideration of how representative were our results. As we responded to the reviewer above in greater detail, the available estimations in the literature about number of individuals who had COVID-19 disease at least once by March 2022 in England are in line with the number of patients included in our study.

Line 237-241: how do these ethnicity compositions compare with other data sources?

We appreciate the reviewer’s interest on how ethnicity is represented with other data sources. This is a large topic to discuss, with a lot of nuances. First, we would like to distinguish data sources within the UK or outside. As expected, ethnicity compositions in data sources outside the UK will be completely different, but we would also like highlight that the specific ethnic groups per se (even those with the same name) can be different/not always comparable. If we focus in the UK, the different nations (Wales, Northern Ireland, Scotland and England) have different ethnicity compositions, where England has the largest diversity. Both, the NHS England and SAIL are representative of the England and Wales populations, respectively: we had previously published a database description paper where we explored ethnicity in the NHS England in terms of completeness, coverage and the granularity, and we confirmed that GPPR (the primary care data set where we identified the participants for the England population in this study) is representative of the population when compared to the ethnicity distribution provided by the ONS 2021 Census [reference 17 in the paper]. Similarly, the Wales data source published a profile of the SAIL data, including details on ethnicity [references 12, 14 and 18 in the paper].

Reference for Ethnic groups in England and Wales - ONS Census 2021:

<https://www.ons.gov.uk/peoplepopulationandcommunity/culturalidentity/ethnicity/bulletins/ethnicgroupenglandandwales/census2021>

NHS England and SAIL references are included in the manuscript (Methods - Data Sources section & Ethnicity classifications section):

14. Wood, A. et al. Linked electronic health records for research on a nationwide cohort of more than 54 million people in England: data resource. *BMJ* 373, n826, doi:10.1136/bmj.n826 (2021).

16. Jones, K. H., Ford, D. V., Thompson, S. & Lyons, R. A. A Profile of the SAIL Databank on the UK Secure Research Platform. *Int J Popul Data Sci* 4, 1134, doi:10.23889/ijpds.v4i2.1134 (2019).

23. Pineda-Moncusi, M. et al. Ethnicity data resource in population-wide health records: completeness, coverage and granularity of diversity. *Sci Data* 11, 221, doi:10.1038/s41597-024-02958-1 (2024).

24. Akbari, A. et al. Exploring ethnicity dynamics in Wales: a longitudinal population-scale linked data study and development of a harmonised ethnicity spine. *BMJ Open* 14, e077675, doi:10.1136/bmjopen-2023-077675 (2024).

Reviewer #2 (Remarks to the Author):

Thank you to the authors for this interesting paper examining ethnic and racial disparities in severe COVID-19 outcomes in England and Wales. This study makes an important contribution by using comprehensive data and granular ethnicity categories to document the continued inequities in COVID-19 outcomes, specifically 28-day mortality and 30-day cardiovascular disease acute event (CVD). They leverage large national datasets to highlight inequities occurring within the larger racial and ethnic categories that are typically utilized in research. The paper has potential, however there are a couple areas of concern that should be addressed: 1) aligning the methods, results, tables, and figures; and 2) the definition, discussion, and interpretation of race, ethnicity and disparities.

We would like to thank the reviewer for the time spent in reviewing our manuscript, which has helped us improving its quality. After replying all the specific comments below, we sincerely hope we have addressed all the points included in 1) and 2).

1) Aligning the methods, results, tables, and figures:

It is hard to follow the comparisons between the different classification systems in the figures. If comparing the results by classification system is a central research question, then it would be helpful to create a figure or table that more clearly shows the differences between the results from different classifications systems.

We acknowledge the complexity of understanding the results across the different ethnic categories and appreciate the reviewer's valuable feedback. Indeed, this is one of the multiple difficulties of using ethnicity is such depth as we provide in the manuscript, but we believe that the provided information is very valuable. We are the first to show what was the impact of COVID-19 mortality and CVD across very specific ethnic groups like Turkish/Turkish Cypriot or Central/South/Latin American, which show a higher incidence of COVID-19 mortality and CVD that was never previously reported.

Following the reviewer's suggestion, we have composed a supplementary figure to help understanding the different classification systems and their impact in the study outcome.

Key revisions now included:

Different Levels of Ethnicity Codes

Supplementary Figure 3. Representation of the three classification systems in the NHS England and the impact of using them. High-level codes are the broader ethnicity categories whilst those SNOMED represent the most granular groups available. Use of NHS or SNOMED codes show differences across different ethnic groups that are masked when using the high-level codes. For instance, Arab men had lower incidence COVID-19 death compared to Other Ethnic Group, whilst incidence of COVID-19 death in the Central/South/Latin America population was more than double. The red dotted line indicates the age-standardised incidence rates of the Other Ethnic group.

2) The data are reported by sex, but no conclusions are discussed in the text. If sex differences represent an important dimension of rate differences, then these results should be reported. If not, then the results may not need to be presented by sex.

We sincerely thank the reviewer for their helpful suggestions. Prior literature regarding COVID-19 mortality show men were more likely to die than women. Additionally, CVD is more common among men than women in the general population. Thus, we decided from the study design phase that sex stratification was essential. We agree with the reviewer that despite this stratification was clearly essential for us, we did not mention its importance in the manuscript. We have now included the following text in the first paragraph of the discussion to state the need for this stratification, and expanded the conclusions and introductions.

Key revisions now included:

Discussion:

“Prior studies reported men had higher risk of COVID-19 mortality when compared to women.^{3-5,32} Additionally, CVD is more prevalent in men than women in the general population.³³ Thus, we analysed the risk of COVID-19 mortality and CVD separately for men and women. Our results confirm men generally had higher incidence and risk of COVID-19 mortality than women. Likewise, men had higher incidence and risk of CVD, except for Middle Eastern women, whose incidence of CVD was higher than Middle Eastern man.”

3. Ramirez-Soto, M. C., Ortega-Caceres, G. & Arroyo-Hernandez, H. Sex differences in COVID-19 fatality rate and risk of death: An analysis in 73 countries, 2020-2021. *Infez Med* 29, 402-407, doi:10.53854/liim-2903-11 (2021).
4. Doerre, A. & Doblhammer, G. The influence of gender on COVID-19 infections and mortality in Germany: Insights from age- and gender-specific modeling of contact rates, infections, and deaths in the early phase of the pandemic. *PLoS One* 17, e0268119, doi:10.1371/journal.pone.0268119 (2022).
5. Fabiao, J. et al. Why do men have worse COVID-19-related outcomes? A systematic review and meta-analysis with sex adjusted for age. *Braz J Med Biol Res* 55, e11711, doi:10.1590/1414-431X2021e11711 (2022).
32. Office-for-National-Statistics. Coronavirus (COVID-19) and the different effects on men and women in the UK, March 2020 to February 2021, <<https://www.ons.gov.uk/peoplepopulationandcommunity/healthandsocialcare/conditionsanddiseases/articles/coronaviruscovid19andthedifferenteffectsonmenandwomenintheukmarch2020tofebruary2021/2021-03-10>> (2021).
33. Bots, S. H., Peters, S. A. E. & Woodward, M. Sex differences in coronary heart disease and stroke mortality: a global assessment of the effect of ageing between 1980 and 2010. *BMJ Global Health* 2, e000298, doi:10.1136/bmjgh-2017-000298 (2017).

Conclusions:

“We confirmed a generally higher risk of COVID-19 mortality and CVD among men than women.”

And Introduction:

“On the other hand, prior studies have associated COVID-19 with many cardiac complications,¹⁰ and an increased mortality risk in people with history of cardiovascular disease (CVD).¹¹ The ONS had reported a higher CVD mortality in ethnic minorities,¹² however, little is known about the differences CVD across ethnic groups after COVID-19 infection.¹³”

10. Basu-Ray I, Almaddah Nk, Adeboye A, et al. Cardiac Manifestations of Coronavirus (COVID-19) [Updated 2024 Feb 12]. In: StatPearls [Internet]. Treasure Island (FL): StatPearls Publishing; 2024 Jan-. Available from: <https://www.ncbi.nlm.nih.gov/books/NBK556152>

11. Singh, A. K. et al. Prevalence of co-morbidities and their association with mortality in patients with COVID-19: A systematic review and meta-analysis. *Diabetes Obes Metab* 22, 1915-1924, doi:10.1111/dom.14124 (2020).

12. Ali, R., Raleigh, V., Majeed, A. & Khunti, K. Life expectancy by ethnic group in England. *BMJ* 375, e068537, doi:10.1136/bmj-2021-068537 (2021).

13. Islam, S. J. et al. County-Level Social Vulnerability is Associated With In-Hospital Death and Major Adverse Cardiovascular Events in Patients Hospitalized With COVID-19: An Analysis of the American Heart Association COVID-19 Cardiovascular Disease Registry. *Circ Cardiovasc Qual Outcomes* 15, e008612, doi:10.1161/CIRCOUTCOMES.121.008612 (2022).

3) The hazard models do not seem like they add much to the paper - they are not discussed in the conclusion and do not seem to show substantively different results from the main models. They also introduce new sources of bias (e.g., time to diagnosis). The authors should consider cutting these results-- this will also help to streamline the results section so it's more concise.

We thank the reviewer for raising the point of what is the added value of including the HR when both analysis, incidence and hazard ratios had the same results. In this line, we had included both in the discussion. We started indicating the direction of the IR and then we confirm these findings with the hazard models as can be read at the beginning of the third paragraph:

“Different ethnic groups in England and Wales had increased incidence of CVD after COVID-19 diagnosis, when compared to White population. The risk was confirmed after adjusting by

confounders in ethnic groups in England, and for Asian women and men with unknown ethnicity in Wales.”

In our PPI sessions and in the meetings with the study expert committee, we were encouraged to add the hazard models in the paper. The added value of the hazard models is that these were adjusted for confounding, and including both, incidence and HR increases the robustness of our study. Time to diagnosis bias is inherent of survival analysis. We have acknowledged residual (unadjusted/unobserved) confounding as a potential limitation in the Strengths and limitations section:

“Finally, we must acknowledge the likelihood that residual (unadjusted/unobserved) confounding could -at least partially- account for the observed differences, which is an inherent limitation of observational studies.”

4) Discussion of race and ethnicity:

One major point is how race and ethnicity are discussed in the article. The authors should be explicit about 1) how they are defining and conceptualizing race and ethnicity and 2) the interpretations and implications of racial and ethnic health disparities. It is great that the authors explicitly state how they categorized each racial and ethnic category (according to the four different classification schemas). The difference between race and ethnicity is not articulated and there is no indication about how race and ethnicity were reported. Were they self-reported/self-defined by the patient? Assigned by the physician? Even more importantly, there is no discussion about how the authors are interpreting racial and/or ethnic disparities in COVID-19 severe outcomes. Are the authors implying that there are underlying biological differences between the race and ethnic groups that make them more likely to have severe COVID-19 outcomes? Or are they implying that there are underlying social and structural causes including racism and classism that produce inequitable conditions and unique vulnerabilities to COVID-19 severe outcomes? Please clarify. Nature offers some guidance on this topic: <https://www.nature.com/articles/d41586-023-00973-7>

We would like to thank the reviewer for pointing out that we need to clarify how ethnicity was defined in the study in order to distinguish it from the concept of race. UK collects self-assigned ethnicity data and not data on race. Indeed, UK has a legal obligation to collect ethnicity data as part of the Public Sector Equality Duty in the Equality Act 2010.

Ethnicity is a concept that comprises different elements, including physical appearance, race, culture, language, religion, nationality and identity elements. We would like to clarify in the text that we are reporting ethnicity, which per definition is self-report by an individual's own.

We previously published how ethnicity data is collected and used in the NHS England data and in SAIL, and we have included both references in the manuscript to allow any interested reader to find more about it without having to over-extend the length of the methods.

We have included the following modifications in the manuscript (within Methods, at the beginning of the Ethnicity classifications section).

Key revisions now included:

“Ethnicity is a multifaceted concept that may encompass various components, including physical appearance, race, culture, language, religion, nationality, and aspects of identity.¹⁷ Ethnicity was self-reported by the patients at the GP or hospital, and the most recent record

was used in England, as defined in Pineda-Moncusí et al. (2024),²³ and in Wales, as defined in Akbari et al. (2024).²⁴”

23. Pineda-Moncusí, M. et al. Ethnicity data resource in population-wide health records: completeness, coverage and granularity of diversity. *Sci Data* 11, 221, doi:10.1038/s41597-024-02958-1 (2024).

24. Akbari, A. et al. Exploring ethnicity dynamics in Wales: a longitudinal population-scale linked data study and development of a harmonised ethnicity spine. *BMJ Open* 14, e077675, doi:10.1136/bmjopen-2023-077675 (2024).

In regards of how to interpret ethnic disparities in COVID-19 severe outcomes, each ethnicity is composed by a group of people that self identifies themselves within a group that shares similitudes with them, which may include similar health outcomes. Even though the causes of having a higher or lower risk for a specific health outcome may be multifactorial, we can still detect differences across various ethnic groups. In other words, while we may not know the cause with complete certainty, we can still identify health inequities.

We have expanded the discussion as requested. The following text has been included at the beginning of the discussion to help contextualising the use of ethnicity as a proxy to measure health disparities across different groups of individuals.

Key revisions also included:

“Ethnicity is a social construct composed of various components, including physical appearance, race, culture, language, religion, nationality, and aspects of identity.²³ Each ethnicity is composed of a group of individuals who identify themselves within a collective that shares similarities with them, which may include similar health outcomes. While the causes of risk differences for a specific health outcome have been shown to be multifactorial,³⁴ we can still identify disparities across ethnic groups.^{35,36}”

23. Pineda-Moncusí, M. et al. Ethnicity data resource in population-wide health records: completeness, coverage and granularity of diversity. *Sci Data* 11, 221, doi:10.1038/s41597-024-02958-1 (2024).

34. Katikireddi, S. V. et al. Unequal impact of the COVID-19 crisis on minority ethnic groups: a framework for understanding and addressing inequalities. *Journal of Epidemiology and Community Health* 75, 970-974, doi:10.1136/jech-2020-216061 (2021).

35. Wang, S. & Mak, H. W. Generational health improvement or decline? Exploring generational differences of British ethnic minorities in six physical health outcomes. *Ethn Health* 25, 1041-1054, doi:10.1080/13557858.2018.1469736 (2020).

36. Eto, F. et al. Ethnic differences in early onset multimorbidity and associations with health service use, long-term prescribing, years of life lost, and mortality: A cross-sectional study using clustering in the UK Clinical Practice Research Datalink. *PLoS Med* 20, e1004300, doi:10.1371/journal.pmed.1004300 (2023).

5) Additional edits

5.1) Occupation (e.g. essential worker vs not) is another potential important sociodemographic characteristic to consider, given substantial occupational segregation by race-ethnicity. If available, the authors should consider including this variable.

Thank you very much for the suggestion. We agree with the reviewer that it would be interesting to see the interaction between ethnicity and occupation. Unfortunately, we do not have occupation recorded in our databases.

5.2) In the discussion, the authors write that the mortality gap disappeared after the appearance of omicron. Further theorizing about this change would be helpful-for example, do the authors believe this change is related to the timing of vaccine introduction, or something to do with the omicron variant?

We would like to thank the reviewer for the interesting question. The COVID-19 vaccine was proven to reduce the number of severe outcomes. The effect of the vaccination can be clearly observed for instance in the decrease of mortality incidence between the 6 months after the vaccination onset [Jul-Dec 2021], and the next 6 months when vaccination was decelerating (given that most people already received the full immunisation [i.e., the two doses]) [Jan-Jun 2022]. However, we can see another drop once Omicron became majoritarian. Thus, we think that the closing of gap in mortality could have been a combination of both factors. However, despite we also see a reduction of the CVD incidence after the vaccine onset, the decrease after Omicron became majoritarian is smaller, and it does not close the CVD gap. We have expanded the discussion as suggested.

Key revisions now included:

“Analysing the risk of mortality over time in England, our survival analysis showed the mortality gap across different ethnic groups disappeared after the appearance of the Omicron variant (period of Jan-April 2022).⁴³ **This could be a result of the vaccination plus the lower severity of the original Omicron variant.** Despite these findings, health disparities in COVID-19 outcomes **were** not over in the Omicron era when analysing CVD risk after SARS-Cov-2 infection.”

5.3) The CVD outcome should be introduced in the introduction, and then further detail on why it is included, and its conceptualization should be added to the methods. For example, how did the authors identify a new CVD diagnosis versus a prior diagnosis?

Thank you for the comment. We decided to include CVD as an outcome of interest for COVID-19 given that cardiac injury is recognized as one of the most frequent complications of the disease. Furthermore, despite it is well known that certain ethnic groups such as Black have higher risk of CVD, we identified there was a gap in the literature where risk of cardiovascular events after SARS-CoV-2 infections across different ethnic groups was not explored.

A new CVD diagnosis was defined as the first CVD recorded event in the clinical history after the SARS-CoV-2 infection. Additionally, we included in the adjustment different risk factors related with CVD, including prior recorded diagnosis before index date of CVD, atrial fibrillation and hypertension.

In addition to the key revisions included in Question 2 from Reviewers 2, we incorporated the rationale of why we included CVD in the study in the introduction. Additionally, we have clarified in Statistical analysis- Survival analysis that the list of adjustment variables for CVD was obtained from a predictive tool for cardiovascular risk that was developed and validated in the UK:

Key revisions now included:

Introduction

“On the other hand, prior studies have associated COVID-19 with many cardiac complications,¹⁰ and an increased mortality risk in people with history of cardiovascular disease (CVD).¹¹ The ONS had reported a higher CVD mortality in ethnic minorities,¹² however, little is known about the differences CVD across ethnic groups after COVID-19 infection.¹³”

10. Basu-Ray I, Almaddah Nk, Adebayo A, et al. Cardiac Manifestations of Coronavirus (COVID-19) [Updated 2024 Feb 12]. In: StatPearls [Internet]. Treasure Island (FL): StatPearls Publishing; 2024 Jan-. Available from: <https://www.ncbi.nlm.nih.gov/books/NBK556152>

11. Singh, A. K. et al. Prevalence of co-morbidities and their association with mortality in patients with COVID-19: A systematic review and meta-analysis. *Diabetes Obes Metab* 22, 1915-1924, doi:10.1111/dom.14124 (2020).
12. Ali, R., Raleigh, V., Majeed, A. & Khunti, K. Life expectancy by ethnic group in England. *BMJ* 375, e068537, doi:10.1136/bmj-2021-068537 (2021).
13. Islam, S. J. et al. County-Level Social Vulnerability is Associated With In-Hospital Death and Major Adverse Cardiovascular Events in Patients Hospitalized With COVID-19: An Analysis of the American Heart Association COVID-19 Cardiovascular Disease Registry. *Circ Cardiovasc Qual Outcomes* 15, e008612, doi:10.1161/CIRCOUTCOMES.121.008612 (2022).

Methods

"The list of covariates included in the adjustment was obtained from a list of key risk factors for addressing confounding in COVID-19 mortality provided by the ONS,²⁸ along with a list of cardiovascular confounders obtained from a predictive tool for cardiovascular risk that was developed and validated in the UK.²⁹"

28. Office-for-National-Statistics. Updating ethnic contrasts in deaths involving the coronavirus (COVID-19), England and Wales: deaths occurring 2 March to 28 July 2020, <<https://www.ons.gov.uk/peoplepopulationandcommunity/birthsdeathsandmarriages/deaths/articles/updatingethnicontrastsindeathsinvolvingthecoronaviruscovid19englandandwales/deathsoccurring2marchto28july2020>> (2021).

29. Hippisley-Cox, J., Coupland, C. & Brindle, P. Development and validation of QRISK3 risk prediction algorithms to estimate future risk of cardiovascular disease: prospective cohort study. *BMJ* 357, j2099, doi:10.1136/bmj.j2099 (2017).

5.4) The authors include a number of covariates in the analysis. It would help to conceptualize why these are included—are they confounders? Minor revisions:

Yes, the covariates in the analysis were included to reduce any potential confounding. We obtain the list of confounders from a prior study conducted by the Office of National Statistics in the UK in COVID-19 patients, and we incorporated a list of cardiovascular confounders obtained from a risk predictive tool called QRISK3 that have been developed and validated in the UK, and have been implemented in clinical practice by the National Health Service (NHS) in the UK.

We have included this information within the Methods, at the end of the Survival analysis section, as suggested.

Key revisions now included:

"The list of covariates included in the adjustment was obtained from a list of key risk factors for addressing confounding in COVID-19 mortality provided by the ONS,²⁸ along with a list of cardiovascular confounders obtained from a predictive tool for cardiovascular risk that was developed and validated in the UK.²⁹"

5.5) It is not clear what the percentages in the second paragraph of the abstract mean – please specify

We thank the reviewer for the observation. We have clarified the abstract that these percentages are the increased risk when compared to White British men and women, respectively.

Key revisions now included:

“...English data show mortality decreased during the Omicron variant's dominance, whilst CVD risk remained elevated for certain ethnic groups when compared to White populations (**increased CVD risk when compared with White British men**: 58% Pakistani, 120% White and Asian, and 17% Any other White background; **and with White British women**: 75% Bangladeshi, 55% Caribbean, and 82% Any Other Ethnic Group).

5.6) The paper states that in the survival analysis the “adjustment variables did not contain missing data” does that mean that a complete case analysis (i.e. excluded those with missing data) was conducted or that the variables had no missing data? Please specify.

We agree the reviewer that the manuscript quality could be improved by clarifying the mentioned statement.

We confirm that we did not restrict the analyses to users with complete variables, which would bias the results. Instead, the variables selected did not include any missing values in the patients' records. All individuals in the analysis had an index of multiple deprivation (IMD) recorded that was provided by the Office of National Statistics in the case of England and by the Wales Demographic Service Dataset (WDS) for Wales. The remaining variables included in the analyses are diagnosis or conditions that are recorded (like the smoking status), record of vaccination and record of pregnancy. The absence of a diagnosis/condition code, vaccination record or pregnancy record in the individual's medical history (which is linked across primary care, hospitals and specific registries such as the pregnancy and vaccination datasets) is considered as the absence of such. Despite there is a limitation that we may be missing diagnoses, such practice is well established when conducting research in Electronic Medical Records.

We have included the limitation of potential missing diagnoses in the limitation section, and we have expanded the sentence the “adjustment variables did not contain missing data” to ensure it won't be ambiguous for readers.

Key revisions now included:

“Adjustment variables did not contain missing data: **individuals with invalid age and sex were excluded (see Participants section), IMD of all patients was provided by the Office of National Statistics for England and by the Wales Demographic Service Dataset for Wales, whilst the remaining variables were based on the presence or absence of its record in the clinical history of the patient (the absence of the code in the individual's record was considered as no condition).**”

“**Additionally, we cannot exclude that possibility that the recording of smoking, diagnosis of conditions, vaccination and pregnancy status might been incomplete.**”

5.7) Using line graphs for the figures that show data over time would make it easier to see the trends.

We thank the reviewer for the suggestion, and we would like to show that we have deeply considered the option to show the data as line graphs. We have included a sample of the plots we produced in the word document containing the reply to the reviewers. As it can be seen in the plots, such option impede us to give the context of how by using the high-level groups researchers are not considering the variability of much specific ethnic groups, which is a message we aim to emphasise in our paper. In addition, despite we agree that the original plots require to spend more time looking at them to understand the results, they can

summarise in one look the differences across the NHS categories within the same high-level categories and vs the other subgroups.

5.8) A clear statement about the gap this article is addressing would be helpful.

Thanks for the suggestion, we have added the following sentence at the beginning of the strengths and limitations section.

Key revisions now included:

“Our study fills a gap in the literature regarding the risk of CVD after SARS-Cov-2 infection across different ethnic groups.”

5.9) Did the authors use direct or indirect age adjustment? Related, some studies in the US have found even wider racial-ethnic disparities in middle ages (e.g. 40-50)- the authors could consider a sensitivity analysis using age bands rather than age adjustment.

Thanks for the consideration. The provided incidence rates were age-standardised, which contains the different age bands that are later weighted by the prevalence of that same band in the population and then merged. Despite we agree that wider disparities could potentially be observed by stratifying by age groups, one of the principal reasons to use this

methodology was to avoid the loss in precision given by the reduced sample size of the age subgroups, especially for small ethnic groups such as Arab or Gypsy/Irish Traveller.

5.10) What is the geography for the IMD? Neighborhood?

The IMD in the UK is obtained using a wider range of indicators including income, employment, education and skills, health and disability, crime, barriers to housing and services, and living environment as described by *Kontopantelis, E. et al. (2018)*. We have included a sentence to clarify how IMD is obtain in the UK.

Key revisions now included:

“The IMD is obtained using a broad spectrum of indicators including income, barriers to housing and services, education and skills, employment, health and disability, crime, and living environment; and it is often categorised into fifths, with 1 denoting the most deprived and 5 the least deprived areas.²¹”

21. Kontopantelis, E. *et al.* Geographical epidemiology of health and overall deprivation in England, its changes and persistence from 2004 to 2015: a longitudinal spatial population study. *Journal of Epidemiology and Community Health* **72**, 140-147, doi:10.1136/jech-2017-209999 (2018).

5.11) It would be helpful to include a sensitivity analysis including those with less than 1 year of primary care records. This can help address potential selection bias around access to primary care.

Thanks for the suggestion. Unfortunately, our access the data is now limited and we cannot re-run additional analyses than the originally planned that were approved by the consortium.

5.12) Would suggest that the authors list the various data sources in a table rather than as a bulleted list.

Thanks for the suggestion, we have replaced the bulleted list of England and Wales data sources for a table (Table 1).

Key revisions now included:

Table 1. Linked data sources included in England and Wales databases

5.13) Figures five and six are quite hard to read because the text is so small and there's so much information in each panel.

We appreciate the reviewer's concern about the readability of figures 5 and 6. The quality of the figures was reduced in the generation of the manuscript proof. We will ensure the final version has the original resolution.

5.14) The figures after Figure 7 do not have titles. Are they supposed to be part of the supplementary materials? If so, it would still be useful to include the titles. It would also be helpful to see a table of contents for the supplementary materials.

We acknowledge and appreciate the reviewer's concern. We had reviewed the pdf generated in the submission and we understood what the reviewer pointed. The 7 figures the reviewer is referring are the same 7 main figures from the manuscript (in its correspondent order) in a larger scale. We would like to assure the reviewer that the supplementary material contains an index to navigate across the 6 figures and 14 tables, and that each figure and table have their correspondent legends.

Reviewer #3 (Remarks to the Author):

We extend our thanks to both reviewers for their careful consideration and for their contributions to improving this manuscript.

Reviewer #4 (Remarks to the Author):

This manuscript “Are ethnic disparities in COVID-19 severe outcomes over? Analysis of 5.3 million individuals in England and Wales from 2020-2022” presents from an analysis of linked electronic health data among adults by ethnic category to compare 30 day risk of mortality and cardiovascular disease. The COVID-19 pandemic highlighted the importance of understanding health disparities with the recognition individuals with high social vulnerability or minority ethnicities were more likely to be infected with SARS-CoV-2 and have severe outcomes from acute COVID-19. These disparities were recognized early in 2020 but it is unclear whether these disparities have persisted. Further, many studies in this area have used broad ethnic categories, limiting the ability to develop more targeted interventions or to understand potential factors contributing to the disparities. These are two reasons why this study provides important information and contributes to our understanding of severe COVID-19 outcomes.

One of the main objectives of this manuscript, as relayed by the title, is to examine how disparities by the outcomes and by ethnic groups may have changed over time. There is extensive literature in this area as to how to measure health disparities, how to compare differences across groups and over time. This manuscript would be strengthened by a review of this literature and re-analyses. For example, figure 5 that reports the incidence by ethnicity group across the time periods, could be presented as trend lines for the incidence of mortality (e.g.) within each ethnic group. The slope and trajectory of each ethnic group line could be compared over time. This would provide information as to whether the incidence of mortality (e.g.) was decreasing more rapidly in one ethnic group compared to another. The authors could also compare the absolute difference in incidence for each group compared to Whites at time period 1 and then compare the absolute difference compared to Whites at the last time period. This would also provide information as to whether or not the gap between mortality by ethnic group is narrowing over time.

We extend our thanks to both reviewers for their careful consideration and for their contributions to improving this manuscript.

Detailed comments for the authors-

1. The title mentions “COVID-19 severe outcomes” but the authors do not include risk of hospitalization for acute COVID-19 or risk of ICU admission as severe outcomes. It is not clear why these are not included.

We are grateful to the reviewers for their thoughtful comment. When designing the study, we considered adding hospitalization and ICU admission. However, due to computational constraints of the environment and the study timelines of the funding (it was a 1-year grant) we need to narrow the number of outcomes in our research. Given that many previous

publications focused of ICU admission, decided to focus on mortality as a benchmark comparison since we had the ONS results to compare, and we included CVD as cardiovascular research is one of the main aims of the CVD-COVID-IMPACT consortium.

2. The authors define severe outcomes as mortality within the first 28 days or cardiovascular disease within the first 30 days after acute COVID-19. There are a number of questions related to the selection of these outcomes. First, it is not clear why the time frame for the two outcomes differs or why 28 days and 30 days were selected. Next, it is not clear why cardiovascular disease (CVD) is included in this paper.

Thank you for the comment. We decided to include CVD as an outcome of interest for COVID-19 given that cardiac injury is recognized as one of the most frequent complications of the disease. Furthermore, despite it is well known that certain ethnic groups such as Black have higher risk of CVD, we identified there was a gap in the literature where risk of cardiovascular events after SARS-CoV-2 infections across different ethnic groups was not explored. When we designed the study, we select 28 days for COVID-18 mortality and 30 days for cardiovascular events based on prior literature using this time windows as plausible windows to attribute COVID-19 mortality and CVD mortality.

We incorporated the rational of why we included CVD in the study in the introduction, as well as in the first paragraph of the discussion:

Key revisions now included:

Introduction

“On the other hand, prior studies have associated COVID-19 with many cardiac complications,¹⁰ and an increased mortality risk in people with history of cardiovascular disease (CVD).¹¹ The ONS had reported a higher CVD mortality in ethnic minorities,¹² however, little is known about the differences CVD across ethnic groups after COVID-19 infection.¹³”

10. Basu-Ray I, Almaddah Nk, Adeboye A, et al. Cardiac Manifestations of Coronavirus (COVID-19) [Updated 2024 Feb 12]. In: StatPearls [Internet]. Treasure Island (FL): StatPearls Publishing; 2024 Jan-. Available from: <https://www.ncbi.nlm.nih.gov/books/NBK556152>

11. Singh, A. K. et al. Prevalence of co-morbidities and their association with mortality in patients with COVID-19: A systematic review and meta-analysis. *Diabetes Obes Metab* 22, 1915-1924, doi:10.1111/dom.14124 (2020).

12. Ali, R., Raleigh, V., Majeed, A. & Khunti, K. Life expectancy by ethnic group in England. *BMJ* 375, e068537, doi:10.1136/bmj-2021-068537 (2021).

13. Islam, S. J. et al. County-Level Social Vulnerability is Associated With In-Hospital Death and Major Adverse Cardiovascular Events in Patients Hospitalized With COVID-19: An Analysis of the American Heart Association COVID-19 Cardiovascular Disease Registry. *Circ Cardiovasc Qual Outcomes* 15, e008612, doi:10.1161/CIRCOUTCOMES.121.008612 (2022).

Discussion

“Prior studies reported men had higher risk of COVID-19 mortality when compared to women.^{3-5,32} Additionally, CVD is more prevalent in men than women in the general population.³³ Thus, we analysed the risk of COVID-19 mortality and CVD separately for men and women. Our results confirm men generally had higher incidence and risk of COVID-19 mortality than women. Likewise, men had higher incidence and risk of CVD, except for Middle Eastern women, whose incidence of CVD was higher than Middle Eastern man.”

3. Ramirez-Soto, M. C., Ortega-Caceres, G. & Arroyo-Hernandez, H. Sex differences in COVID-19 fatality rate and risk of death: An analysis in 73 countries, 2020-2021. *Infez Med* 29, 402-407, doi:10.53854/liim-2903-11 (2021).
4. Doerre, A. & Doblhammer, G. The influence of gender on COVID-19 infections and mortality in Germany: Insights from age- and gender-specific modeling of contact rates, infections, and deaths in the early phase of the pandemic. *PLoS One* 17, e0268119, doi:10.1371/journal.pone.0268119 (2022).
5. Fabiao, J. et al. Why do men have worse COVID-19-related outcomes? A systematic review and meta-analysis with sex adjusted for age. *Braz J Med Biol Res* 55, e11711, doi:10.1590/1414-431X2021e11711 (2022).
32. Office-for-National-Statistics. Coronavirus (COVID-19) and the different effects on men and women in the UK, March 2020 to February 2021, <<https://www.ons.gov.uk/peoplepopulationandcommunity/healthandsocialcare/conditionsanddiseases/articles/coronaviruscovid19andthedifferenteffectsonmenandwomenintheukmarch2020tofebruary2021/2021-03-10>> (2021).
33. Bots, S. H., Peters, S. A. E. & Woodward, M. Sex differences in coronary heart disease and stroke mortality: a global assessment of the effect of ageing between 1980 and 2010. *BMJ Global Health* 2, e000298, doi:10.1136/bmjgh-2017-000298 (2017).

3. The outcome of CVD could be a separate paper. This would also help focus and streamline the results.

We are grateful to the reviewers for their thoughtful comment. In our study, the inclusion of both outcomes COVID-19 mortality and CVD across the different ethnic groups provides new insights. Our results confirm the gap closure in mortality described the ONS while exploring a more granular ethnic groups for England. Additionally, we used the same methodology to explore the differences in CVD across ethnic groups, which were little explored before. Including a known effect such as mortality in this paper helps validating the methodology later used for CVD. We agree with the reviewer that our paper opens the opportunity for new research about CVD after COVID-19 disease and, as included in the conclusions, further studies are required to explain this remaining increased risk of CVD for certain ethnic group of patients diagnosed from COVID-19.

4. Abstract (lines 41-42) and repeated in the conclusions- “COVID-19 mortality and CVD risk was increased in most non-White ethnic groups in England and Asian population in Wales during 2.5 years after the pandemic outbreak.” This sentence could be misinterpreted to imply that the authors focused on outcomes 2.5 years after acute COVID-19. Recommend revising to more accurately report the results”...30-day mortality and 28 day CVD risk was higher among non-White ethnic groups...”

We thank the reviewers for pointing this. We have reworded the sentence in the abstract as suggested.

Key revisions now included:

“COVID-19 **28-day** mortality and **30-day** CVD risk was increased in most non-White ethnic groups in England and Asian population in Wales during the 2.5 years after the pandemic outbreak”

5. Reporting both the NHS and the SNOMED-CT ethnic groups is a valuable contribution of this analysis, but there are a lot of results. Recommend streamlining the paper by focusing only on the NHS ethnic groups, in addition to the 6-census groups in England and 10-level classification for Wales. This would reduce the number of panels for each figure. The authors can focus the results and discussion around these groupings. The results from the SNOMED-CT groups can be presented in the supplemental tables and then briefly discussed as to whether or not they add to this discussion on disparities.

We acknowledge the complexity of understanding the results across the different ethnic categories and appreciate the reviewer's valuable feedback. Indeed, this is one of the multiple difficulties of using ethnicity is such depth as we provide in the manuscript, but we believe that the provided information is very valuable and should be included as main findings. We are the first to show what was the impact of COVID-19 mortality and CVD across very specific ethnic groups like Turkish/Turkish Cypriot or Central/South/Latin American, which show a higher incidence of COVID-19 mortality and CVD that was never previously reported.

Following the reviewer's suggestion, we have composed a supplementary figure to help understanding the different classification systems and their impact in the study outcome.

Key revisions now included:

Supplementary Figure 3. Representation of the three classification systems in the NHS England and the impact of using them. High-level codes are the broader ethnicity categories whilst those SNOMED represent the most granular groups available. Use of NHS or SNOMED codes show differences across different ethnic groups that are masked when using the high-level codes. For instance, Arab men had lower incidence COVID-19 death compared to Other Ethnic Group, whilst incidence of COVID-19 death in the Central/South/Latin America population was more than double. The red dotted line indicates the age-standardised incidence rates of the Other Ethnic group.

6. Severity of acute COVID-19 (hospitalized, ICU, etc) and acute COVID-19 treatment are associated with increased risk of 30 day mortality. These clinical considerations need to be compared across ethnic groups and accounted for in adjusted analysis.

Thank you for this observation. The list of covariates included in the analysis was obtained from a list of key risk factors for addressing confounding in COVID-19 mortality provided by the ONS, along with a list of cardiovascular confounders obtained from a predictive tool for cardiovascular risk that was developed and validated in the UK. Given the extensive methodological research the ONS undertook to adjust for confounding in COVID-19 mortality at 28 days (like we did), severity should be considered when implementing the key risk factor list along with treatment. However, we acknowledge the reviewers point and have include the following text in the strength and limitations section - second paragraph.

Key revisions now included:

“These results must be interpreted taking into account that the diagnosis of COVID-19 in these data did not include cases of lateral flow test (LFT)-only positive COVID-19 cases.³⁸ Moreover, due to limited capacity, testing was restricted outside of secondary care settings during the first wave, meaning diagnoses were clinical, resulting in an increased risk of potential misclassification during that period.³⁹ The selected adjusting factors for COVID-19 mortality were obtained from the ONS,²⁶ which instead of including severity (such as hospitalisation or intensive care unit admission) and specific medication of the acute COVID-19 was not included in the adjustment, included health status before the COVID-19 diagnosis. Other factors may play a role in the interpretation of this results, such as health-seeking behaviour and barriers to accessing health care,⁴⁰ which may exacerbate the differences between the White and non-White study population, where the observed non-White could be more populated by those who are experiencing worse outcomes.”

7. Footnote in the figure states that vaccination status is included in the adjusted variables. How was vaccination defined? Was this based on evidence of any vaccination? Vaccination at least 14 days prior to COVID-19? Then in the survival analysis section of the methods, vaccination status is not included in the list of adjusted variables. Please confirm whether or not vaccination status was included in the models.

Thank you for pointing this. We did adjust for vaccination, but we missed to add it in the covariates section. We have included vaccination in Covariates, within the clinical characteristics.

Key revisions now included:

“record of first dose of COVID-19 vaccination before index date”, as well as “prior COVID-19 vaccination” in the survival analysis section.

8. Results: Did you observe differences in the demographic (age, sex) or clinical (comorbidities, acute COVID-19 severity) across ethnic groups over time? In other words, did the case mix of COVID-19 patients differ over time by ethnic group? Were there differences over time in England compared to Wales?

We thank the reviewers for suggesting this interesting point. We did not include in the analysis a comparison of the patients’ characteristics over time, but we will consider it for future work.

9. Line 263-266 could be moved to the methods.

We acknowledge and appreciate the reviewers’ suggestion. Although this could fit well in methods as the reviewers suggested, we would like to keep it in the results for two main reasons: 1) this was a decision after running the analysis and based on the results obtained; and 2) if explains the logic followed when interpreting the results while the reader are looking at them, which we think it may be easier for them.

10. There are strong reasons to stratify all the results by male, female. This reasoning needs to be added to both the introduction and the discussion.

We thank for pointing this out. Following their advice, we have included the rational in the introduction, and in the first paragraph of the discussion.

Key revisions now included:

In the introduction:

“On the other hand, prior studies have associated COVID-19 with many cardiac complications,¹⁰ and an increased mortality risk in people with history of cardiovascular disease (CVD).¹¹ The ONS had reported a higher CVD mortality in ethnic minorities,¹² however, little is known about the differences CVD across ethnic groups after COVID-19 infection.¹³”

10. Basu-Ray I, Almaddah Nk, Adeboye A, et al. Cardiac Manifestations of Coronavirus (COVID-19) [Updated 2024 Feb 12]. In: StatPearls [Internet]. Treasure Island (FL): StatPearls Publishing; 2024 Jan-. Available from: <https://www.ncbi.nlm.nih.gov/books/NBK556152>

11. Singh, A. K. et al. Prevalence of co-morbidities and their association with mortality in patients with COVID-19: A systematic review and meta-analysis. *Diabetes Obes Metab* 22, 1915-1924, doi:10.1111/dom.14124 (2020).

12. Ali, R., Raleigh, V., Majeed, A. & Khunti, K. Life expectancy by ethnic group in England. *BMJ* 375, e068537, doi:10.1136/bmj-2021-068537 (2021).

13. Islam, S. J. et al. County-Level Social Vulnerability is Associated With In-Hospital Death and Major Adverse Cardiovascular Events in Patients Hospitalized With COVID-19: An Analysis of the American Heart Association COVID-19 Cardiovascular Disease Registry. *Circ Cardiovasc Qual Outcomes* 15, e008612, doi:10.1161/CIRCOUTCOMES.121.008612 (2022).

In the discussion:

“Prior studies reported men had higher risk of COVID-19 mortality when compared to women.^{3-5,32} Additionally, CVD is more prevalent in men than women in the general population.³³ Thus, we analysed the risk of COVID-19 mortality and CVD separately for men and women. Our results confirm men generally had higher incidence and risk of COVID-19 mortality than women. Likewise, men had higher incidence and risk of CVD, except for Middle Eastern women, whose incidence of CVD was higher than Middle Eastern man.”

3. Ramirez-Soto, M. C., Ortega-Caceres, G. & Arroyo-Hernandez, H. Sex differences in COVID-19 fatality rate and risk of death: An analysis in 73 countries, 2020-2021. *Infez Med* 29, 402-407, doi:10.53854/liim-2903-11 (2021).

4. Doerre, A. & Doblhammer, G. The influence of gender on COVID-19 infections and mortality in Germany: Insights from age- and gender-specific modeling of contact rates, infections, and deaths in the early phase of the pandemic. *PLoS One* 17, e0268119, doi:10.1371/journal.pone.0268119 (2022).

5. Fabiao, J. et al. Why do men have worse COVID-19-related outcomes? A systematic review and meta-analysis with sex adjusted for age. *Braz J Med Biol Res* 55, e11711, doi:10.1590/1414-431X2021e11711 (2022).

32. Office-for-National-Statistics. Coronavirus (COVID-19) and the different effects on men and women in the UK, March 2020 to February 2021, <<https://www.ons.gov.uk/peoplepopulationandcommunity/healthandsocialcare/conditionsanddiseases/articles/coronaviruscovid19andthedifferenteffectsonmenandwomenintheukmarch2020tofebruary2021/2021-03-10>> (2021).

33. Bots, S. H., Peters, S. A. E. & Woodward, M. Sex differences in coronary heart disease and stroke mortality: a global assessment of the effect of ageing between 1980 and 2010. *BMJ Global Health* 2, e000298, doi:10.1136/bmjgh-2017-000298 (2017).

11. Line 398, the first sentence should clarify that the ethnic groups had increased mortality compared to Whites. Please review entire manuscript to make sure it is clear who the comparison group is.

Thank you for the suggestion, we have included the clarification as requested.

Key revisions now included:

“In line with previous estimates reported by the ONS,⁶ most of non-White ethnic groups had an increased mortality **when compared with White populations** in England and Wales.”

12. The discussion would be strengthened by including more details about ethnic disparities in general in England and Wales, whether or not they were present prior to

COVID pandemic, how the pandemic did or did not exacerbate them, and potential reasons why they exist.

We thank the reviewers for valuable feedback. Ethnic disparities in the UK were present before the pandemic as can be read in the Marmot Review from February 2020 [<https://www.health.org.uk/reports-and-analysis/reports/health-equity-in-england-the-marmot-review-10-years-on-0>]. From prior unpublished work in NHS England regarding inequity in access of surgery across ethnic groups, we saw lower representation of minorities having surgery, but this inequity seemed to be maintained rather than exacerbated after the pandemic in the England system. Thus, we think that the pandemic highlighted a problem that was already there in the UK (however, this may be different for other outcomes, and will likely differ completely in other countries). From the literature, there are several papers discussing inequities in the UK across different ethnic groups. Such as Bécares, J. (2013) that already emphasized how Pakistani and Bangladeshi women had mortality rates 10% higher than white women between 1991 and 2011; or Sheena Asthana et al (2018) review of Inequity in cardiovascular care in the NHS for England, which demonstrated that patterns of use of the health care system varied by ethnicity.

We have included the following text in the discussion as suggested.

Key revisions now included:

“Ethnicity is a social construct composed of various components, including physical appearance, race, culture, language, religion, nationality, and aspects of identity.²³ Each ethnicity is composed of a group of individuals who identify themselves within a collective that shares similarities with them, which may include similar health outcomes. While the causes of risk differences for a specific health outcome have been shown to be multifactorial,³⁴ we can still identify disparities across ethnic groups.^{35,36} For instance, we can find multiple examples of research based in the UK showing outcomes such mortality, access to health care or quality of life had been disproportionate across ethnic groups already before the COVID-19 pandemic.³⁷⁻⁴⁰ And when the pandemic struck, reports about how minority ethnic groups were severely more impacted by COVID-19 were disclosed at an earlier time, not long after the initial months.”

23. Pineda-Moncusi, M. et al. Ethnicity data resource in population-wide health records: completeness, coverage and granularity of diversity. *Sci Data* 11, 221, doi:10.1038/s41597-024-02958-1 (2024).

34. Katikireddi, S. V. et al. Unequal impact of the COVID-19 crisis on minority ethnic groups: a framework for understanding and addressing inequalities. *Journal of Epidemiology and Community Health* 75, 970-974, doi:10.1136/jech-2020-216061 (2021).

35. Wang, S. & Mak, H. W. Generational health improvement or decline? Exploring generational differences of British ethnic minorities in six physical health outcomes. *Ethn Health* 25, 1041-1054, doi:10.1080/13557858.2018.1469736 (2020).

36. Eto, F. et al. Ethnic differences in early onset multimorbidity and associations with health service use, long-term prescribing, years of life lost, and mortality: A cross-sectional study using clustering in the UK Clinical Practice Research Datalink. *PLoS Med* 20, e1004300, doi:10.1371/journal.pmed.1004300 (2023).

37. Bécares L (2013) Which Ethnic Groups Have the Poorest Health? *Ethnic Health Inequalities 1991 to 2011*. 10.2307/j.ctt1t89504.14.

38. Matthews D. The impact of ethnicity on health inequalities. *Nurs Times*. 2015 Oct 28-Nov 3;111(44):18-20. PMID: 26665383.

39. Asthana S, Moon G, Gibson A, Bailey T, Hewson P, Dibben C. Inequity in cardiovascular care in the English National Health Service (NHS): a scoping review of the literature. *Health Soc Care Community*. 2018 May;26(3):259-272. doi: 10.1111/hsc.12384. Epub 2016 Oct 16. PMID: 27747961.

40. Watkinson, Ruth Elizabeth et al. 'Ethnic inequalities in health-related quality of life among older adults in England: secondary analysis of a national cross-sectional survey.' *The Lancet Public Health*, Volume 6, Issue 3, e145 - e154

13. Line 469 states that reasons for these disparities are complex and intersectional, but these concepts and background are not presented in this manuscript.

We sincerely appreciate the reviewers' insightful comment. We start the introduction stating that "Health inequity is multifaceted and often underpinned by a complex interplay of determinants, including but not limited to race and ethnicity, sex, and socioeconomic status." And we have now expanded the introduction of the discussion to explain the complexity within the ethnicity definition. We hope both sections will help addressing the request of more background to back up the concepts of complexity and intersectionality related with health inequity and ethnicity.

Key revisions now included:

"Ethnicity is a social construct composed of various components, including physical appearance, race, culture, language, religion, nationality, and aspects of identity.²³ Each ethnicity is composed of a group of individuals who identify themselves within a collective that shares similarities with them, which may include similar health outcomes. While the causes of risk differences for a specific health outcome have been shown to be multifactorial,³⁴ we can still identify disparities across ethnic groups.^{35,36}"

23. Pineda-Moncusi, M. et al. Ethnicity data resource in population-wide health records: completeness, coverage and granularity of diversity. *Sci Data* 11, 221, doi:10.1038/s41597-024-02958-1 (2024).

34. Katikireddi, S. V. et al. Unequal impact of the COVID-19 crisis on minority ethnic groups: a framework for understanding and addressing inequalities. *Journal of Epidemiology and Community Health* 75, 970-974, doi:10.1136/jech-2020-216061 (2021).

35. Wang, S. & Mak, H. W. Generational health improvement or decline? Exploring generational differences of British ethnic minorities in six physical health outcomes. *Ethn Health* 25, 1041-1054, doi:10.1080/13557858.2018.1469736 (2020).

36. Eto, F. et al. Ethnic differences in early onset multimorbidity and associations with health service use, long-term prescribing, years of life lost, and mortality: A cross-sectional study using clustering in the UK Clinical Practice Research Datalink. *PLoS Med* 20, e1004300, doi:10.1371/journal.pmed.1004300 (2023).

14. Figures reporting incidence- add a footnote with details on the method and age groups for age standardization.

We would like to thank the reviewers for the suggestion. We added the following clarification to figures 1, 4 and 5, and supplementary figures 3 and 4.

Key revisions now included:

"To estimate the age-standardised incidence rates, age-specific incidence rates were calculated for 5-year age bands and then combined using the 2013 European Standard Population weights from 30 to 90+ age groups."

15. Figure 1 (e.g.) consider reporting only high level and NHS ethnic groups for England. This would allow the figure to be reduced to a and b- one for England and one for Wales, with 2 panels each -one for mortality and one for CVD.

We acknowledge the complexity of understanding the results across the different ethnic categories. Indeed, this is one of the multiple difficulties of using ethnicity is such depth as we provide in the manuscript. However, we belief that the provided information is very valuable. We are the first to show what was the impact of COVID-19 mortality and CVD

across very specific ethnic groups like Turkish/Turkish Cypriot or Central/South/Latin American, which show a higher incidence of COVID-19 mortality and CVD that was never previously reported.

We have added a new supplementary figure to help understanding the different classification systems and their impact in the study outcome.

Key revisions now included:

“Supplementary Figure 3. Representation of the three classification systems in the NHS England and the impact of using them. High-level codes are the broader ethnicity categories whilst those SNOMED represent the most granular groups available. Use of NHS or SNOMED codes show differences across different ethnic groups that are masked when using the high-level codes. For instance, Arab men had lower incidence COVID-19 death compared to Other Ethnic Group, whilst incidence of COVID-19 death in the Central/South/Latin America population was more than double. The red dotted line indicates the age-standardised incidence rates of the Other Ethnic group.”

16. Figure 2 (e.g.- applies to all HR figures) HR needs to start at 1.0 and then graph the estimate below or above 1.0. Consider also reporting the results on a log scale.

We acknowledge and appreciate the reviewers' feedback, and we would like to respectfully explain our disagreement with the current request. HR may have values below 1 (which we can see in our results) and therefore the plots need to allow when there is a lower risk than the comparison group (ie., a HR value below 1.0).

RESPONSE TO REVIEWERS' COMMENTS

Authors' note: We would like to thank all the reviewers for their time and effort that had helped us improving the quality of our manuscript, and the opportunity to address once more their inquiries and suggestions.

Reviewer #1 (Remarks to the Author):

I have focused this review on the responses to my initial comments.

In response to my first comment expressing concern about the descriptive nature of the research, the authors say: "To address such disparities, those responsible for delivering health care and regulators need to adapt the guidelines and health policies, only then an analysis to confirm that such guidelines and policy changes had addressed the disparities could be performed." I find this too vague to be useful and am concerned that the impact of the work will be limited.

We acknowledge the reviewer's point of view regarding the impact of the current work. Whilst we consider that the first step to tackle inequity in health care across different ethnicities is to detect disparities in health outcomes such as the CVD gap in patients diagnosed with COVID-19 from different ethnic groups, we as researchers cannot implement changes in the current clinical practice but we can provide the evidence to implement such changes, which needs to be led by health care authorities and healthcare providers. Hence, and in response to the reviewer's first comment, we added the following sentence to the manuscript: 'The analysis we undertook shows where the gaps on COVID-19 mortality and CVD are and where healthcare providers can focus to address them.'

Despite we cannot change clinical practice, we as researchers can keep an eye on the CVD disparities regarding COVID-19 and keep highlighting this gap until is closed. We added the following sentence at the end of the conclusions to emphasize it.

Key revisions now included:

"Further studies are recommended to monitor the ongoing disparities in CVD outcomes among COVID-19 patients from different ethnic groups."

My comment #2. I am confused by the authors' response. They state that there were over 21 million cases in the UK by 1st April 2022. They then say the number of individuals who had ever had COVID-19 was estimated to be 212,000 in Wales and 4.1 million in England by the same date. Are they saying that *on average* a person having COVID-19 at least once had COVID-19 five times within two years? That would certainly be worthy of comment if true.

We thank the reviewer for their comment and apologise for not providing a clearer response in the prior comment. The 21.4 million cumulative cases reported by the Johns Hopkins COVID-19 statistics refer to the entire UK, which includes Northern Ireland and Scotland—areas not covered in our analysis. However, we acknowledge that, of the 67.79 million people estimated as UK population in 2022, around 57.1 million would correspond to England. The UK source data was obtained from the UKHSA data dashboard, which no longer displays data prior to April 2024. Therefore, we cannot assess the accuracy of the reported 21 million UK cases (e.g., whether duplicates occurred, such as individuals submitting multiple positive tests for the same episode when self-testing at home). Hence,

we also provided information from the Welsh Government, which reported the number of individuals who had COVID-19 at least once—rather than cumulative case counts—to inform the reviewer. However, to simplify the discussion, our key revisions focused on providing evidence in the manuscript that both NHS England and SAIL are highly representative of the populations of England and Wales.

My comment #4: The authors have clarified what they did and who told them to do that, but have not addressed this part of my comment “If the latter, the results are critically based on what happens to outcomes in the reference category over time. This is not discussed in the manuscript.” If the rates in the reference category are changing over time, this is important context for the changes over time in the relative differences.

We would like to thank the reviewer for their clarification. In the overall hazard ratios (HR), the enrolment period [i.e., the time of COVID-19 diagnosis] was added in the adjustment, using the first period —23 Jan 2020 to 30 Jun 2020 — as the reference (specified in the supplementary tables reporting the overall HR results). In the stratification by enrolment period, the results are based on the ethnicity reference category (i.e., White or White British ethnic groups).

While we agree with the reviewer that the rates in the reference category changed over time (as shown in the incidence rate plots), the focus of our analysis was to illustrate the increased risk of the other ethnic groups when compared to the White/White British population within the same time period.

Responses to specific comments

These are generally fine, but I remain unclear on these:

My comment: “Abstract - Doesn’t start with a statement of what we know and what we don’t know and therefore the knowledge gap that this paper fills is not clear”. Everyone faces the same word constraints. I don’t think there is a need for additional space.

We acknowledge the reviewer’s feedback and have revised the abstract to clarify the specific gaps in the literature that this paper addresses.

Key revisions now included:

“Previous studies reported higher COVID-19 mortality risk across ethnic groups, but data on ethnic disparities in COVID-19-related cardiovascular disease (CVD) were lacking.”

My comment: “Line 227: detailed of how models were adjusted for “deprivation index” are not given”. The authors have told me WHAT deprivation measure they used, but not HOW they have adjusted for it. It is a continuous but not cardinal variable.

Thank you very much for raising this. In the R script it is possible to let the model know the form of the variable. For IMD, which is distributed on a scale of 1 to 5, it was treated by the model as an ordinal variable, not treated as continuous. We have added the clarification in the manuscript.

Key revisions now included:

“The IMD is obtained using a broad spectrum of indicators including income, barriers to housing and services, education and skills, employment, health and disability, crime, and living environment; and it is often categorised into fifths, with 1 denoting the most deprived

and 5 the least deprived areas.³⁰ **The IMD was treated as an ordinal variable by the model.**”

Minor point: there are several typos in the responses and revised manuscript text.

We would like to thank the reviewer for pointing this out. We have revised the manuscript to identify and correct typographical errors, with particular focus on the text added in response to previous comments.

Reviewer #2 (Remarks to the Author):

Thank you for addressing our comments and questions. The clarity of the manuscript is much improved. We have a couple remaining comments below.

2) The data are reported by sex, but no conclusions are discussed in the text. If sex differences represent an important dimension of rate differences, then these results should be reported. If not, then the results may not need to be presented by sex.

a. Follow up comment: I appreciate the additional context and justification added by the authors. It would also be valuable to highlight that while overall men have had higher rates over covid 19 mortality that is not true in all geographies or subgroups. It is useful to highlight the places or subgroups where women had higher rates (if applicable). Where you say “men generally had higher incidence...” you could also list the subgroups where women had higher incidence.

b. See the following citation: Danielsen AC, Lee KM, Boulicault M, Rushovich T, Gompers A, Tarrant A, Reiches M, Shattuck-Heidorn H, Miratrix LW, Richardson SS. Sex disparities in COVID-19 outcomes in the United States: Quantifying and contextualizing variation. Soc Sci Med. 2022 Feb;294:114716. doi: 10.1016/j.socscimed.2022.114716. Epub 2022 Jan 10. PMID: 35042136; PMCID: PMC8743486.

We would like to thank the reviewer for the comment and appreciate the suggestion. Following their advice, we have added the provided reference in the first paragraph of the introduction, where we mention that sex, race and ethnicity are health determinants that are multifaceted and a complex interplay. We have also included the reference in the Discussion section, in the sentence: “Prior studies reported men had higher risk of COVID-19 mortality when compared to women.” Additionally, we reviewed instances where women showed higher mortality rates than men in our results and have added a few examples in the Discussion.

Key revisions now included:

“Our results confirm men generally had higher incidence and risk of COVID-19 mortality than women, **with fewer exceptions for specific ethnic groups where women’s hazard rates were higher, such as Chinese women during the period of Jul-Dec 2020, or African and ‘White and Black Caribbean’ women during the period of Jul-Dec 2021.**”

Reference 6: Danielsen AC, Lee KM, Boulicault M, Rushovich T, Gompers A, Tarrant A, Reiches M, Shattuck-Heidorn H, Miratrix LW, Richardson SS. Sex disparities in COVID-19 outcomes in the United States: Quantifying and contextualizing variation. Soc Sci Med. 2022 Feb;294:114716. doi: 10.1016/j.socscimed.2022.114716. Epub 2022 Jan 10. PMID: 35042136; PMCID: PMC8743486

5.1) What is the geography for the IMD? Neighborhood?

a. Follow up comment: Thank you for providing additional information about the IMD. In the statement that you added (“The IMD is obtained using a broad spectrum of

indicators including income, barriers to housing and services, education and skills, employment, health and disability, crime, and living environment; and it is often categorised into fifths, with 1 denoting the most deprived and 5 the least deprived areas”), how is the area referenced defined? Is it neighborhoods, or another administrative or political geographic unit?

We would like to thank the reviewer for their interest and appreciate the opportunity to clarify this point. The index of multiple deprivation (IMD) is the most widely used index in England to measure the relative deprivation in small areas in England called lower-layer super output areas (LSOA). IMD ranks each small area in England from most deprived (rank 1) to least deprived (rank 32,844 in 2019) based on the combination of seven factors and their respective weights.

Full details of how this measure was calculated are available in the NHS England website and in the 'English indices of deprivation 2019: research report'

Links:

<https://www.england.nhs.uk/about/equality/equality-hub/national-healthcare-inequalities-improvement-programme/what-are-healthcare-inequalities/deprivation/>

<https://www.gov.uk/government/publications/english-indices-of-deprivation-2019-research-report>

We have clarified our statement in the report. We have included the above references in case any reader would like to know more about how this index is calculated in England.

Key revisions now included:

“The **IMD is calculated for each LSOA in England to rank the most deprived areas (rank 1) to the least deprived, and is based on a combination of seven indicators that cover a broad spectrum**,^{31,32} including income, barriers to housing and services, education and skills, employment, health and disability, crime, and living environment; and it is often categorised into fifths, with 1 denoting the most deprived and 5 the least deprived areas”

Reviewer #3 (Remarks to the Author):

We would like to thank the reviewer for their time and implication in reviewing our manuscript.

Reviewer #4 (Remarks to the Author):

The authors have carefully addressed each reviewer's comments.

While the gap that this research fills is clearer now, and the focus on cardiovascular disease is clearer, the title of the paper should be revised. The refers to severe outcomes, but the analysis is focused on mortality and CVD. I recommend revising the title of the paper to reflect this.

We would like to thank the reviewers for their recognition and encouraging comments, and for this last recommendation. Following their advice, we updated the title to the following:

Key revisions now included:

Title:

"Ethnic disparities in COVID-19 mortality and cardiovascular disease in England and Wales between 2020-2022"